# Structurally Modified MXenes-Based Catalysts for Application in Hydrogen Evolution Reaction: A Review

**Raja Rafidah Raja Sulaiman** [1,†], **Abdul Hanan** [2,†], **Wai Yin Wong** [1,*], **Rozan Mohamad Yunus** [1], **Kee Shyuan Loh** [1], **Rashmi Walvekar** [3], **Vishal Chaudhary** [4] and **Mohammad Khalid** [2,*]

1  Fuel Cell Institute, Universiti Kebangsaan Malaysia, Bangi 43600, Selangor, Malaysia
2  Graphene & Advanced 2D Materials Research Group (GAMRG), School of Engineering and Technology, Sunway University, No. 5, Jalan Universiti, Bandar Sunway, Petaling Jaya 47500, Selangor, Malaysia
3  Department of Chemical Engineering, School of New Energy and Chemical Engineering, Xiamen University Malaysia, Jalan Sunsuria, Bandar Sunsuria, Sepang 43900, Selangor, Malaysia
4  Research Cell & Department of Physics, Bhagini Nivedita College, University of Delhi, Delhi 110043, India
*  Correspondence: waiyin.wong@ukm.edu.my (W.Y.W.); khalids@sunway.edu.my (M.K.)
†  These authors contributed equally to this work.

**Abstract:** Green hydrogen production via electrocatalytic water splitting paves the way for renewable, clean, and sustainable hydrogen ($H_2$) generation. $H_2$ gas is produced from the cathodic hydrogen evolution reaction (HER), where the reaction is catalyzed primarily from Pt-based catalysts under both acidic and alkaline environments. Lowering the loading of Pt and the search for alternative active catalysts for HER is still an ongoing challenge. Two-dimensional MXenes are effective supports to stabilize and homogenously distribute HER-active electrocatalysts to boost the HER performance. Factors involved in the effectiveness of MXenes for their role in HER include transition metal types and termination groups. Recently, tailoring the conditions during the synthesis of MXenes has made it possible to tune the morphology of MXenes from multilayers to few layers (delaminated), formation of porous MXenes, and those with unique crumpled and rolled structures. Changing the morphology of MXenes alters the surface area, exposed active sites and accessibility of electrolyte materials/ions to these active sites. This review provides insight into the effects of varying morphology of MXenes towards the electrocatalytic HER activity of the MXene itself and MXene composites/hybrids with HER-active catalysts. Synthesis methods to obtain the different MXene morphologies are also summarized.

**Keywords:** hydrogen; hydrogen evolution reaction; nanomaterials; transition metals; MXene

## 1. Introduction

Hydrogen ($H_2$) as a fuel offers the pathway for a clean, abundant, renewable and efficient source of energy that is currently crucial in the worldwide effort to lower emissions of harmful pollutants. The implementation of $H_2$ in fuel cells came about around 219 years after its discovery, as can be seen in the timeline in Figure 1. The technology has evolved from its first use in the U.S. Apollo Space Program in the 1950s to the development of the first fuel-cell vehicle in the 2000s [1]. Throughout this course, the utilization of $H_2$ as fuels further developed for their production using renewable energy approaches since the implementation of the world's first solar-powered $H_2$ production towards the development of the grid system for hydrogen generation in the 2020s and onwards. Hydrogen is an abundant element and non-toxic. Utilizing $H_2$ in hydrogen fuel-cell systems is able to generate electricity with only water and heat as by-products [2,3]. However, the majority (95%) of global hydrogen production is still based on non-renewable resources via steam methane reforming, as of 2020, emitting 830 tonnes of $CO_2$ annually [4,5]. Therefore, sustainable hydrogen generation should be sourced from renewable sources while reducing

$CO_2$ emissions. Green hydrogen adopts $H_2$ production methods that do not rely on non-renewable sources such as fossil fuels. Water electrolysers are an emerging technology that utilizes photocatalytic or electrocatalytic water splitting to generate $H_2$ [6–8]. Integrating electrolysers with renewable sources such as solar and wind power provides us with green and environmentally friendly $H_2$ sources with no toxic by-products. An alkaline water electrolyser is a developed technology that generates $H_2$ by water electrolysis. However, highly concentrated potassium hydroxide (KOH) solution in alkaline electrolysers poses a great risk of component corrosion [9]. The proton exchange membrane water electrolyser (PEMWE) and anion exchange membrane water electrolyser (AEMWE) offer a more compact design that uses a lower-concentration electrolyte (0.5 M sulfuric acids ($H_2SO_4$) for PEMWE and 1 M KOH for AEMWE) and is able to overcome some limitations related to corrosion and carbonate formation in conventional electrolysers [9–13].

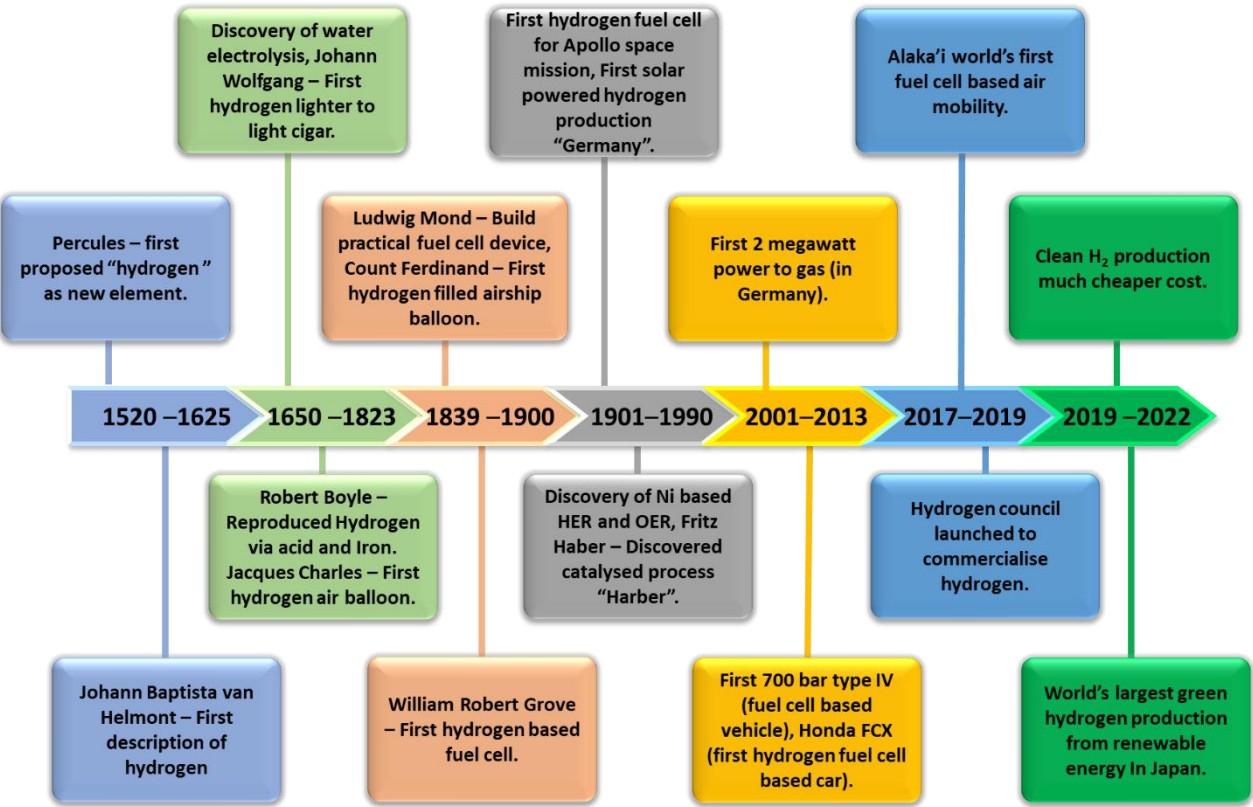

**Figure 1.** Timeline on the development of $H_2$ utilization as fuel.

The electrochemical water-splitting system has two essential reactions: hydrogen evolution reaction (HER) and oxygen evolution reaction (OER). The ideal thermodynamic potential to split water into its component hydrogen and oxygen is 1.23 V. In practice, the potential is much higher due to the mass, electrolyte and transport resistances, in addition to the slow HER and OER kinetics. Additional potential to achieve water splitting is known as overpotential [14]. HER is the cathode reaction in the water electrolyser where $H_2$ is generated. Active research is still ongoing in the field of HER. Moreover, increasing publications related to HER show that challenges and opportunities still exist to be tackled (Figure 2). The rate at which HER occurs varies between the acidic and alkaline electrolyte environments [15,16]. The $H_2$ production rate is determined from the activity of the HER electrocatalyst, where it is desired to have a high current density at the lowest possible potential. Therefore, the smaller HER overpotential of an electrocatalyst indicates its high activity. The benchmark HER electrocatalysts are based on noble or platinum-group metals (PGM), most commonly Pt/C, owing to their very low overpotential in acidic and alkaline

conditions [17]. Therefore, to lower the cost of electrode fabrication, PGM-based catalyst loading must be minimized without losing the optimum HER activity.

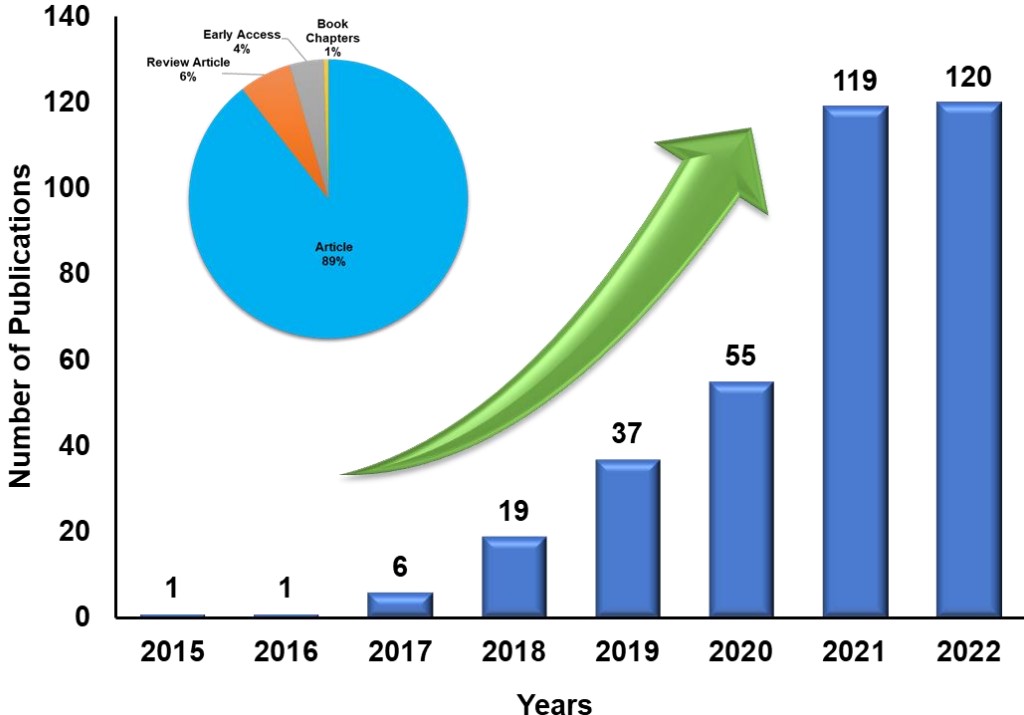

**Figure 2.** Various research publications during the last ten years toward HER (keywords: HER, MXenes). Source: Web of Science, https://www.webofscience.com/wos/woscc/summary/aa561b6 b-209c-40c7-9557-00595d867918-62e97e6f/relevance/1, accessed on 14 November 2022.

Various electrocatalysts alternative to Pt/C has been extensively studied to identify the best lower-cost catalysts for HER [18]. Ni, Mo and Co-based materials are among the potential catalysts for HER. This includes alloys such as NiMo, transition-metal sulphides, chalcogenides and nitrides [19]. Single-atom catalysts (SACs) such as those of Pt and Ir SACs offer the advantage of utilizing ultra-low loadings of PGM-based catalysts [20]. However, nanoparticles/nanosheets of these catalysts, including SACs, are prone to aggregation, causing instability in their HER performance. Two-dimensional (2D) nanomaterials such as graphene and MXene are excellent supports to stabilize these catalysts, which in turn also enhances the HER. MXenes, belonging to a family of 2D transition metal carbides/nitrides, contain an abundance of the termination groups on their basal planes that enable them to anchor catalyst nanoparticles and SACs, facilitating the catalysts' distribution [20–23]. These termination groups (–O, –F and –OH) also participate in HER as active sites. Their high conductivity helps lower the electron transfer resistance for enhancing intrinsic HER activity. The types of transition metal and whether it is carbide or nitride determine the electronic structure of MXene and its effectiveness towards HER alongside the termination groups [21]. Past reviews have highlighted the properties of different types of MXenes, the effect of termination groups, their synthesis and their roles as supports for catalysts for HER as well as other applications, including OER [21], photocatalytic water splitting [24], $CO_2$ reduction [24,25] and nitrogen reduction reaction [26]. While the MXene types and their atomic structure significantly affect catalytic activity, the Mxene's morphology also contributes to the catalysts. Since its discovery in 2011, MXenes with different structures and morphologies, such as multilayer, few-layer and porous, have been successfully synthesized and investigated. Changing morphologies can determine how well the MXene participates in the HER by the ease of access to its active sites, its resistance to restacking, overall surface area, effectiveness in supporting HER-active phases and its durability. Hence, identifying the most suitable out of all the possible MXene morphologies can help

cater to the improvement of the overall HER performance. To the best of our knowledge, the effects of changing the morphology of MXene towards HER performance of MXene and MXene-supported electrocatalysts have yet to be reviewed.

This review highlights different morphologies of MXenes and their effects on the electrocatalytic HER in water splitting. The well-known morphologies, which are the multilayer and few-layer MXenes, followed by several types of porous MXenes as well as specially structured MXenes comprising crumpled/rolled MXenes, are elaborated in the following sections. Synthesis methods to fabricate these different morphologies are also provided. Finally, brief statements on the remaining challenges and prospects highlight the opportunities to be further explored in the study of MXene's morphological influence on HER.

## 2. Hydrogen Evolution Reaction

HER occurs as a cathode reaction within the water electrolyser system, where water is reduced to generate $H_2$ ($2H^+ + 2e^- \rightarrow H_2$) [27]. It is a two-electron transfer system with one catalytic step in general [28–30]. Therefore, to achieve high kinetic effectiveness for electrochemical water splitting, an active electrocatalyst must be used to reduce the overpotential that drives the HER process [31]. The PGM-based electrocatalyst platinum (Pt) is a well-known HER catalyst that needs smaller overpotentials even at high reaction rates (especially in acidic solutions). Pt/C can typically exhibit overpotentials as low as 20 mV to around 80 mV at 10 mA/cm$^2$ [32,33]. However, the scarcity and high cost of Pt limit its technological usage, prompting the effort to minimize the loading of Pt in electrodes or replace it with lower-cost transition-metal-based electrocatalysts.

### 2.1. Mechanism of Electrochemical HER

The mechanism of HER is dependent on the driving environment, such as alkaline and acidic solutions [34], represented by Equations (1) and (2), respectively. Further, the literature shows that these equations are furthermore divided into various sub-steps [35]. It contains proposed HER kinetics in acidic and alkaline environments, as depicted in Figure 3.

$$2H_3O + 2e^- \rightarrow H_2 \uparrow \qquad \text{(Acidic)} \qquad (1)$$

$$2H_3O + 2e^- \rightarrow H_2 \uparrow +2OH^- \qquad \text{(Alkaline)} \qquad (2)$$

Three possible systems for HER in an acidic environment:

$$H_3O^+ + e^- \rightarrow H_{ads} + H_2O \qquad \text{(Volmer)} \qquad (3)$$

$$H_{ads} + H_3O^+ + e^- \rightarrow H_2 + H_2O \qquad \text{(Heyrovsky)} \qquad (4)$$

$$H_{ads} + H_{ads} \rightarrow H_2 \qquad \text{(Tafel)} \qquad (5)$$

Three possible systems for HER in an alkaline environment:

$$2H_2O + 2e^- \rightarrow 2H_{ads} + 2OH^- \qquad \text{(Volmer)} \qquad (6)$$

$$H_{ads} + H_2O + e^- \rightarrow H_2 + OH^- \qquad \text{(Heyrovsky)} \qquad (7)$$

$$H_{ads} + H_{ads} \rightarrow H_2 \qquad \text{(Tafel)} \qquad (8)$$

where $H_{ads}$ is a hydrogen atom adsorbed on an active site. Normally, one stage actively limits the electrochemical response and is known as the rate-determining step (rds) [36]. Hydrogen evolution energy is firmly reliant upon the terminal material, such as a mercury (Hg) electrode, which shows slow energy, while the HER on platinum is one of the quickest known electrocatalytic processes [37]. It is striking that the energy is dependent upon varieties of boundaries, such as the nature of the electrolyte or the crystalline nature and direction of the electrode (single-crystalline, polycrystalline, amorphous and so on) [38].

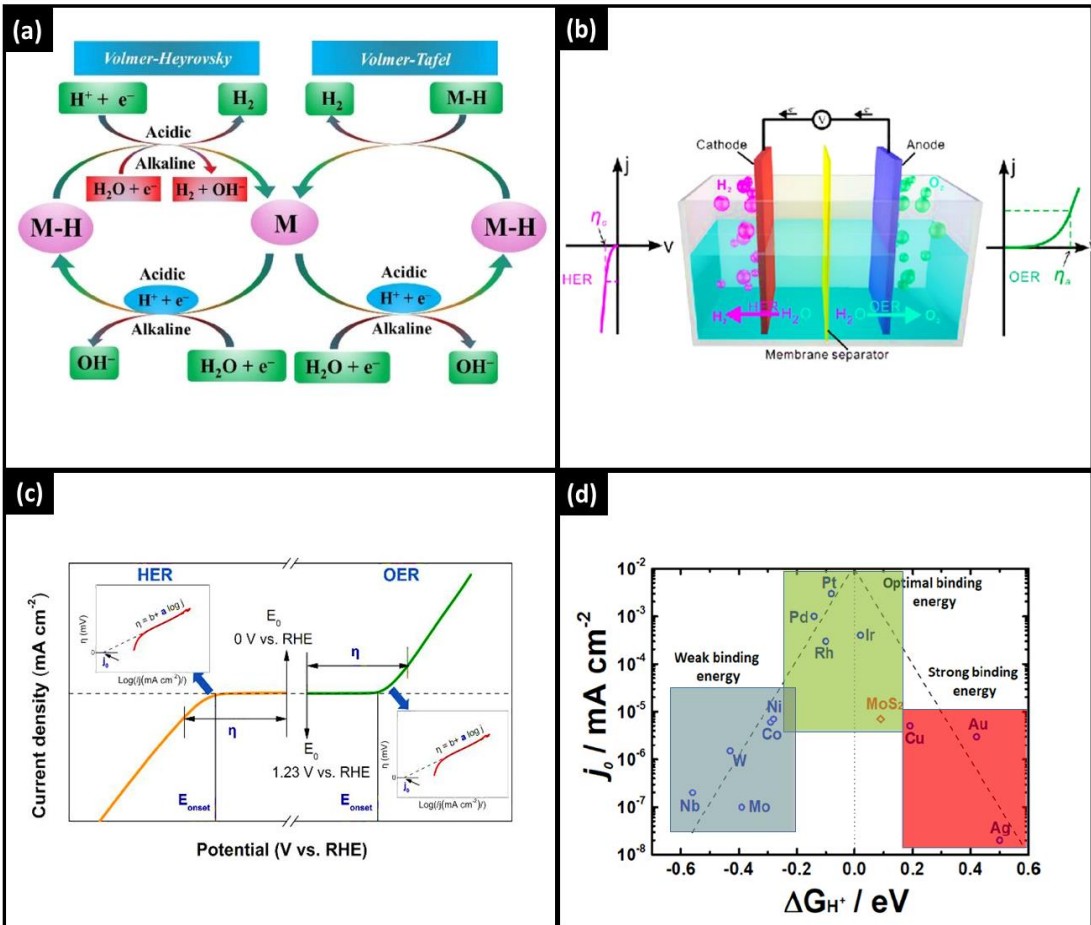

**Figure 3.** (**a**) The Volmer, Heyrovsky and Tafel reaction mechanisms of electrochemical water splitting. (**b**) Electrocatalysis chamber with anodic and cathodic approaches. (**c**) Overall, HER and OER kinetics. (**d**) Various electrocatalysts for HER through volcanic plot.

However, HER is faster in acidic than alkaline environments because of the water dissociation half-cell reaction at the HER (cathode) of electrolysers under an alkaline one [39]. When water electrolysis is performed in an acidic medium, hydronium ions ($H_3O^+$) are reduced to vapor dihydrogen ($H_2$) [40]. Such sub-electrode reactions would take place at the voltage of such a reference hydrogen electrode from a thermodynamic perspective (RHE). Equation (3) describes the first stage of this reaction, which is the reduction of a proton on such an active site of the surface of the catalyst, proceeded either by the evolution of molecule $H_2$ (Equation (4)) or through the recombination of two adsorbed protons (Equation (5)). In addition, the overall Tafel, Heyrovsky and Volmer steps are depicted in Figure 3a. The whole electrocatalysis chamber, which demonstrates the HER and OER, is schematically presented in Figure 3b.

Extensive efforts have been made to identify alternative catalysts to Pt/C that are electrocatalytically active in alkaline and acid conditions with exceptional long-term durability. Several studies have attempted to lower the loading of Pt on electrodes while maintaining the HER activity. Conversely, transition metal (TM)-based HER electrocatalysts such as nickel and molybdenum-based nanoparticles/nanosheets have shown excellent potential as lower-cost catalyst material. To date, the activity of TM-based catalysts is still inferior to Pt-based catalysts, yet some are found to have extended durability than the latter. TM-based nanoparticles/nanosheets can be susceptible to aggregation, negatively affecting the HER. Two-dimensional (2D) materials such as graphene, $MoS_2$ and MXene are actively studied as support for these nanoparticles/nanosheet electrocatalysts to minimize their aggregation as well as participate in the HER reaction to boost the activity. MXenes show

the advantage of high electron-conducting properties and availability of HER-active sites, making them attractive for application in electrocatalytic HER.

### 2.2. Catalyst Activity toward Overpotential, Current Density and Tafel Slope

Because of fundamental activity, hurdles found on both the anode and the cathode are really what primarily causes the excessive potential, also known as overpotential ($\eta$), to exist. Therefore, assessing electrocatalysts' activity and overpotential is a significant feature. The overpotential value associated with a current density of 10 mA/cm$^2$ is typically utilized to compare the activity of various catalysts [41–43].

The Tafel slope as well as exchange current, which are also additional parameters to assess activity through overpotential vs. reactive current connection, are expressed in the following equation: $\eta = a + b \log j$, where $j$ is the current density, and $\eta$ is the overpotential (shown in Figure 3c). The linear connection refers to two notable kinetic parameters for the Tafel plot. The other is the exchange current density ($j_0$), which may be determined by extracting the current at zero overpotential. One is the Tafel slope ($b$). According to the kinetics of electron transport, the Tafel slope ($b$) is associated with the catalytic reaction mechanism [44]. The lower Tafel slope indicates that the electrocatalytic reaction kinetics is occurring more quickly and that the overpotential shift results in a significant increase in current density [45]. Under equilibrium conditions, overall basic electron transfer is described by the exchange current density [46]. Greater charge transfer rates and a lower response barrier are correlated with increased exchange current density.

### 2.3. Catalyst Activity for Current–Time Curve

Stability is a key factor in determining if a catalyst has the potential to be used in experimental water-splitting cells [47]. There are two approaches to determining stability. One of those is by using chronoamperometry (I-t curve) and chronopotentiometry (E-t curve), which measure both occurrences with time under a constant potential or the potential variation with time under a fixed current [48]. The higher the stability of the catalyst, the faster the tested current or potential is the same. People frequently set a current density of greater than 10 mA/cm$^2$ for at least 10 h of testing in order to compare results with those of other research groups [35]. Another method is cyclic voltammetry (CV), which determines current by cycling the potential and often requires more than 5000 cycles at a scan rate (such as 50 mV/s) [49].

The chronopotentiometric technique used linear sweep voltammetry (LSV) to investigate an overpotential change before and after the durability test. The electrocatalyst with the lowest potential change is considered to be the desirable electrocatalyst [50].

### 2.4. Efficiency toward Turnover Frequency (TOF) and Faradaic Efficiency

Turnover frequency (TOF) is an important parameter for describing the kinetics rate of catalytic sites, which indicates significant activity of the metal catalysts [51]. In addition, the TOF generally shows how several reactants may be transformed into the required product per active site per unit of time [52]. Furthermore, calculating the total TOF value for more heterogeneous electrocatalysts for catalytic sites at every electrode is often an estimation [53]. Furthermore, while being an imperfect method, TOF is a crucial tool for comparing the catalytic activity of diverse catalysts, particularly within a comparable system or under similar conditions [54].

Its faradaic efficiency is a quantitative technique for defining the effectiveness of transferring electrons from an external circuit toward the surface of the electrode for such an electrochemical reaction [55]. The ratio of experimentally examined amount of H$_2$ or O$_2$ to theoretically determined mass of H$_2$ or O$_2$ is known as faradaic efficiency [56–58]. The theoretical values can be estimated using chronoamperometry or chronopotentiometric analysis. On the other hand, the experimental values can be obtained by measuring the gas generation using the water-gas displacement technique or gas chromatography [59].

The focus of research and development affects the study of electrocatalyst stability, activity and efficiency. Furthermore, in accordance with the specific concentration for efficiency, analysis, structural characterization and process determination, the current studies of reaction, efficiency and stability may be gathered in three areas for the synthesis and production of an electrocatalyst [60]. Assessment of the current/potential-time curve, on the other hand, provides information for assessing the stability of the electrocatalyst, which is helpful for practical applications [61]. Finally, estimating overpotential, Tafel slope, exchange current density, faradaic efficiency and turnover frequency are the primary parameters for assessing electrocatalytic kinetics [62]. Notably, coupling these electrochemical approaches to spectroscopic and microscopic levels provides the structural properties required to design a robust and active electrocatalyst.

## 3. State-of-Art HER Electrocatalysts

### 3.1. Noble-Metal-Based Electrocatalysts

Noble metals, especially Pt-group metals that include PGMs such as Pt, Pd, Ir, and Ru, have the best catalytic performance toward HER [63]. Pt is seen at the top of the volcanic curve in Figure 3d. Furthermore, the practical deployment of these metals has still been limited by their high cost and unavailability. To find a remedy, a catalyst design that makes sense with minimal metal load and high metal efficiency is necessary [64].

Pt can be substantially better-used when alloyed with transition metals, and these alloys' synergistic effects may change the electronic environments, which would significantly boost the HER electroactivity [65]. With an ultralow loading Pt concentration of 7.7%, Sun et al. reported the in situ fabrication of a nanoscale alloy [66] at the current density of 10 mA/cm$^2$ (20 percent). It demonstrates longer stability with 90 h of catalytic activity, which is significant. It is reasonable to explain the strong HER efficiency of the PtNi-Ni NA/CC to the downshift of the d-band core of Pt, thereby decreasing the energy barrier of oxygenated species (OH*) on the atomic surface of Pt.

#### 3.1.1. Pt-Based Electrocatalysts

The capacity to change the metal content on the surface is essential for improving the electrocatalytic performance of a variety of HER electrocatalysts based on Pt [67]. Before this, Markovic et al. revealed how to properly construct Ni (OH)$_2$ clusters on a Pt electrode surface, improving HER activity by a magnitude of eight when compared to Pt [68]. H$_2$ is produced when the absorbed hydrogen precursors interact [69]. Huang et al. [70] reported the synthesis of surface-engineered PtNi-O nanostructures with an enhanced NiO/PtNi interface inspired by the synergetic effects between Ni (OH)$_2$ and Pt (111). In alkaline media, this interface structure changes to Ni (OH)$_2$, resulting in a surface interface resembling Ni (OH)$_2$/Pt (111). With 5.1 g$_{pt}$/cm$^2$ of Pt loaded, the electrocatalyst showed lower overpotential values of 39.8 mV at 10 mA/cm$^2$ for HER. Meanwhile, compared with the mass activity of a 70 mV overpotential to a reversible hydrogen electrode (RHE), PtNi-O/C exhibits the maximum activity (7.33 mA/g$_{pt}$), as opposed to 5.35 mA/g$_{pt}$ for PtNi [71].

#### 3.1.2. Ru-Based Electrocatalysts

As alternatives to Pt, bimetallic catalysts have been thoroughly investigated. Due to its lower cost (roughly half that of Pt metal) and comparable hydrogen-moderate H* adsorption energy to Pt, Ru is also another potential candidate for Pt for such HER [72–74]. However, it has been noted that the interaction between Ru nanoclusters, Ru-O-Mo complexes, with oxygen-vacancy-enriched MoO$_2$ would provide the electrocatalytic developed good catalytic performance. Recently, in a basic electrolyte, an overpotential value of 29 mV at 10 mA/cm$^2$ was produced [75]. Furthermore, a Ru-NMCNs-T electrocatalytic activity toward HER only requires the overpotential value of 28 mV and a modest Tafel slope of 35.2 mA/dec to obtain a current density value of 10 mA/cm$^2$ in 1 M KOH [74]. Over the HER, Ru electro-deposition onto NiCCF was studied at room temperature and also across

the range of temperature of 20–50 °C using 0.1 M NaOH, for cathodic overpotential value from −100 to −300 mV vs. RHE [76]. Ru@RuO$_2$ nanorods revealed high polyfunctional catalytic performance with strong stability toward OER and HER at a desired current density value of 10 mA/cm$^2$ with an overpotential value toward OER and HER of 320 mV and 137 mV, respectively [77]. In Ru@Co/N-CNTs, Ru-based nanoclusters on Co/N-doped CNs exhibit significant HER activity for both acidic as well as alkaline conditions [78]. Additionally, Ru/C-TiO$_2$ was produced in a two-way technique, with a mean size of 3.6 nm for Ru. Having an overpotential value of 44 mV for HER in alkaline media, it exceeds Ru/C (107 mV), Pt/(84 mV), and C-TiO$_2$'s minimal performance [79].

### 3.1.3. Ir-Based Electrocatalysts

The melting point and density of iridium (Ir) are quite high. The substance with the greatest corrosion resistance is iridium. The main function of iridium is to make platinum (Pt) harder. It effectively conducts heat and electricity [80–82]. However, various nanocomposites and nanostructures based on Ir elements have been reported for effective electrochemical application due to their improved electrical conductivity, which is an increasingly fast charge mechanism with lower resistance toward electrocatalytic performance. Recently, applications involving catalysis and corrosive environments have been particularly attractive for iridium-nickel (Ir-Ni) film. On copper (Cu) foam, Ir-Ni thin films were electrodeposited using galvanostatic deposition like an electrocatalyst toward the HER [83], higher than that of pristine Ir as well as Ni nanosheets. They successfully attained a current density value of 10 mA/cm$^2$ and an overpotential value of 60 mV with a Tafel slope value of 40 mV/dec. Furthermore, it has been shown that 2D siloxene with functional groups on the surface of Si-H and Si-OH may assist the spontaneously depositing catalytic iridium nanoparticles (Ir NPs) with enhancing HER kinetics [84]. With a loading of 2.1 wt% Ir, the Ir NPs/siloxene catalyst displays considerably better HER activity, achieving the current density value of 10 mA cm$^2$ equivalent to Pt/C (20 wt% Pt). Compared with the given metal Rh electrocatalyst (having the overpotential value of 300 mV as well as a Tafel slope value of 190 mV/dec), the fundamental electrocatalysts are Rh@Pt (having the overpotential value of 139 mV with a Tafel slope of 84.8 mV/dec) and Rh@Ir (having overpotential of 169 mV with Tafel slope of 112 mV/dec) [85]. This recent study shows that significant lattice strain levels are produced at the hetero interface when Ir and Co are combined, creating an active surface area for better reactivity [82]. By increasing the steps of hydrogen adsorption and water separation significantly, the additive impact among heterogeneous nanostructures could significantly improve the HER mechanism, with low overpotential values of 44 mV in an acidic medium, 49 mV for alkaline conditions and 64 mV in a neutral environment at a current density value of 10 mA/cm$^2$. Most recently [86], Ir/N-rGO exhibits exceptional HER efficacy, requiring just 76 and 260 mV overpotential values, respectively, to reach the current density value of 10 mA/cm$^2$ for HER and OER.

### 3.2. Non-Noble-Metal-Based Electrocatalysts

Figure 4 illustrates the various types of non-noble-metal-based electrocatalysts towards HER, including transition-metal oxides (TMO), transition-metal sulphides (TMS), transition-metal carbides (TMC), transition-metal phosphides (TMP) and transition-metal dichalcogenides.

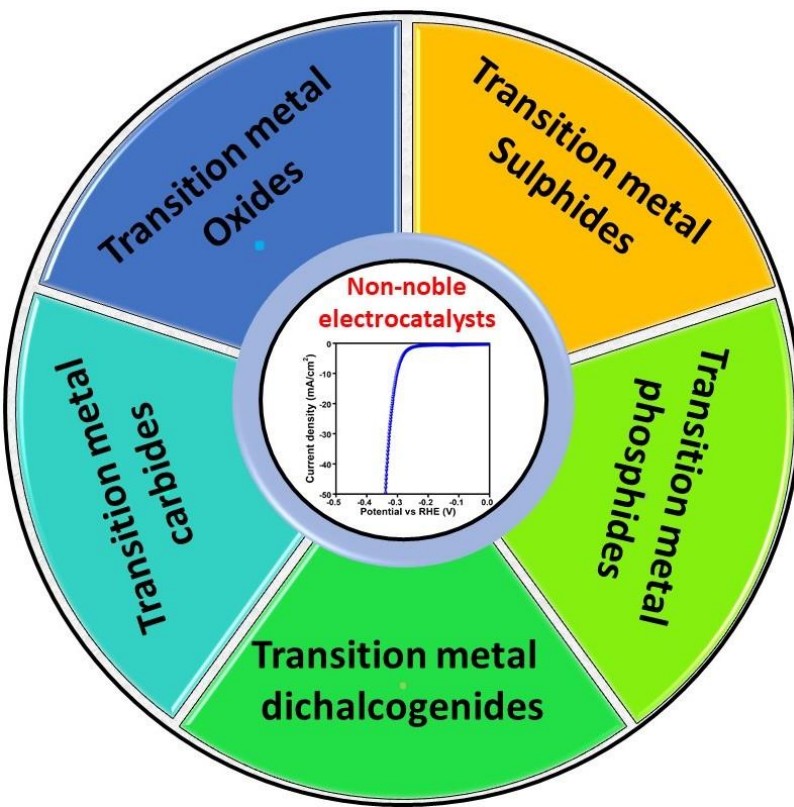

**Figure 4.** Illustration for various non-noble metal-based electrocatalysts toward HER.

### 3.2.1. Transition-Metal Oxides

Transition-metal oxides (TMOs) are often used because of their dielectric material qualities and catalytic properties. Different TMO-based materials have also been reported as efficient electrocatalysts for HER. In this review, we are discussing some individual TMOs based on their composition. The $Mn_3N_2$/PdO displays HER activities [87], with a modest Tafel slope value of 49.6 mV/dec and minimal overpotential of 44.6 mV vs. RHE, to achieve a current density value of 10 mA/cm$^2$. The binary oxides of Gd, In, and Zn demonstrated excellent general activity toward the water-splitting system, with an overpotential of 271 mV at +/−10 mA/cm$^2$ being noted [88]. A ternary oxide composed of $In_2O_3$, ZnO, and $Co_3O_4$ [89] was studied on effective OER as well as HER. The overpotential values of 398 and 510 mV were reported for OER and HER studies at +/−10 mA/cm$^2$ current density value. With a low overpotential value of 101 mV and a modest Tafel slope value of 36 mV/dec, the Ni-Co-P plate shows a strong activity towards HER [90]. $MnO_2$-$TiO_2$ nano-urchins assisted by graphene oxide (GO) [91] exhibit significantly reduced overpotential values of 190 mV and Tafel slope value of 45 mV/dec. A current density of 10 mA/cm$^2$ has been reported by Co non-noble metal-based electrocatalysts Ni-P hollowed nanospheres embedded over reduced graphene oxide (RGO) nanosheets (Co-Ni-P/RGO) for an overpotential value of 207 mV [92]. Regarding limited bi-functional electrocatalytic water-splitting applications, a hybrid cobalt (Co) with nickel (Ni) oxide nanostructure has been reported [93]. The overpotential values were 203 and 378 mV (with a current density value of +/−10 mA/cm$^2$). OER has Tafel slope values of 87 mV/dec and 90 mV/dec.

### 3.2.2. Transition-Metal Sulphides

The transition-metal sulphides are among the several TMs and are crucial for the electrochemical method of producing hydrogen. $CuCo_2S_4$ is synthesized as more than just a low-cost electrocatalyst for water electrolysis to produce hydrogen. In the case of a $CuCo_2S_4$ microsphere with a Tafel slope value of 121 mV/dec, its minimum onset and overpotential values were found to be 61 and 79 mV, respectively [94]. Compared to

2D panels of molybdenum sulphide nanostructure produced on Mo foil, molybdenum sulphide nanosheets exhibit improved HER performance with a lower onset overpotential value of 120 mV as well as a Tafel slope value of 46 mV/dec [95]. Interestingly, for ZCS@NFL/NF [96] to produce a 10 mA/cm$^2$ current density value toward HER in 1 M KOH, a low overpotential value of 110 mV is required. Such standard current densities at 10 mA/cm$^2$ having overpotential values of $-173$, $-208$ and $-283$ mV, determining Tafel slope values of 109.81, 65.92 and 102.06 mV/dec, accordingly, could be achieved by MoS$_2$ nanocrystals, MoSe$_2$ nano-flowers and MoTe$_2$ nanotubes [97]. Having the current density value of 10 mA/cm$^2$, MoS$_2$-encased NiS$_2$ nanohybrids display HER [98] with astonishingly modest overpotential values in both acidic (152 mV) and alkaline (141 mV) environments. The Ni$_3$S$_2$/NC$_2$O hybrid material [99] displayed superior performance with overpotential values of 174, 199 and 249 mV in 0.5 M H$_2$SO$_4$, 1.0 M KOH and 1.0 M phosphate buffer solution (PBS), with a current density of 10 mA/cm$^2$. When compared to the reversible hydrogen electrode (RHE) once at a current density value of 10 mA/cm$^2$ and a Tafel slope of 64 mV/dec, Cu$_3$SnS$_4$-rGO displayed an extremely lower overpotential value of 190 mV [100]. A novel catalytic material with a facet-controlled hollow Rh$_3$Pb$_2$S$_2$ nanocage exhibits high efficacy with an overpotential of 87.3 mV at a current density of 10 mA/cm$^2$ and durability against HER in severe acidic environments [101]. In another approach toward HER in 1 M KOH, the NiFeS/NF electrocatalysts displayed a modest overpotential value of 180 mV [102].

### 3.2.3. Transition-Metal Carbides (TMCs)

Due to their extreme hardness, refractory behavior and affordability, early transition-metal carbides (TMCs) have been widely employed in cutting tools. However, these materials also possess attractive thermal catalytic and electrocatalytic properties. TMCs provide strong charge transfer resistance and sheet resistance as well. Considering the dynamic catalytic activity, typical TMC SiC sheets or nanoparticles (NPs) have a larger potential. Similarly, Mo$_2$C and WC have demonstrated strong HER catalytic activity. Further, an excellent combination is considered the main contributor to high HER activity in the hydrogen adsorption, electrical properties, and d-band electronic density rate (comparable to that of Pt).

Secondly, by hydrothermal method developing carbides in situ on a graphene nanoribbon template and then calcining those at a higher temperature, molybdenum carbide (Mo$_2$C) nanoparticles (anchored to GNRs) were produced. In all alkaline, acidic and neutral conditions, the synthesized carbide material demonstrates efficient electrocatalytic activity and durability. Furthermore, due to the multiple conductive channels for fast electron transport and the broad available surface area containing increased active sites provided by this in situ carbide formation, catalytic activity is improved in all conditions, including acidic, alkaline, and neutral ones.

### 3.2.4. Transition-Metal Phosphides

Transition-metal phosphides (TMPs) as non-precious metal compounds have been focused as heterogeneous electrocatalysts for overall water-splitting systems. The findings of transition-metal phosphides (TMP) are one of the quick-growing sections in developing electrocatalysts with efficiency and stability in both acidic and alkaline mediums. These compounds have been reported as effective electrocatalytical materials toward HER [103]. Such improvements include different approaches, such as the stoichiometric ratio of phosphorous, strange-atom doping/alloying, nano-carbon formation, etc. It is assumed that P atoms have a vital influence on TMP due to their having better conductivity and a unique electronic structure.

Among various TMPs, Ni$_2$P catalysts are one of the best practical catalysts toward HER dates. Liu and Rodriguiz introduced one of the essential electrocatalysts by density functional theory (DFT) [104]. The primary reason Ni2P exhibits better activity over bulk Pt and Ni is due to the hydrogen generated in the HER process, which forms strong bonds with metal (Pt, Ni).

### 3.2.5. Transition-Metal Dichalcogenides

Transition-metal dichalcogenides (TMDs) have received a lot of considerable interest, and they are extensively studied for their potential applications in Li-ion batteries, photovoltaic systems, catalysts for HER, transistors, DNA recognition, photodetectors, memory devices and so on. TMDs are two-dimensional (2D) semiconductor materials with unique mechanical, electrical and optical properties. In next-generation power electronics applications, TMDs can substitute for graphene (metallic substance) and hexagonal boron nitride (hBN, insulator). Through various strategies, such as phase modulation, growth morphology control, site doping and heterostructure preparation, the TMDs have demonstrated improved electrocatalytic performance. Since the observed in-plane resistance of 2D TMDs is lower than the resistance through the basal planes, electrons could be able to move through the basal plane quickly and can reach catalytic sites at the edges.

Except for graphene, TMDs (such as WS$_2$, MoS$_2$, WSe$_2$ and so on) are composed of a 'sandwich'-like structure consisting of a transition-metal layer (such as W, Nb, Mo, etc.) among two layers of chalcogen (such as Se, Te, S, etc.). Moreover, TMDs-based electrocatalysts with 2D layered structures have a bigger surface area and more HER reaction sites.

Atomic covalent bonds to the MoS$_2$ edge have free energy comparable to Pt, according to a 2005 DFT estimate by Norskov et al. [105]. According to this discovery, MoS$_2$ is a promising electrocatalyst for HER. Nils et al. [106] created triangular MoS$_2$ nanocrystals on an Au (111) substrate in various sizes to better pinpoint the real active location of the MoS$_2$ structure. They have shown that the amount of edge sites in the MoS$_2$ designed to work has a linear impact on HER activity. Different methods, such as customizing nanostructures and shapes, have been suggested to reveal the active areas and increase HER performance. In addition, Xie et al. [104] described a method for engineering deficiencies in MoS$_2$ ultrathin nanosheets, which was proven to significantly enhance the electrocatalytic HER efficacy of MoS$_2$. This increased reactivity was related to the rich-defect material's extra active edge sites that were generated by partly cracking the catalytically inactive basal plane. CoS$_2$ was effectively produced with controlled film, microwire and even nanowire morphologies in distinct research by Jin et al. [107].

In summary, transition-metal-based electrocatalysts are more abundant and lower cost, yet their HER performance still does not surpass those of PGM-based electrocatalysts [19,108]. A summary of the HER performance of the catalysts is shown in Table 1. Transition metals such as Fe, Co, Ni, Cu, Mo and W may be utilized to fabricate HER catalysts. Miles and Thomason [109] reported the catalytic efficacy of non-noble metallic-based electrocatalysts in this order: Ni > Mo > Co > W > Fe > Cu. An optimum electrocatalyst for HER during an alkaline medium is reported to be metallic Ni, while metallic Co-based catalysts were also studied for HER electrolysis [110–112]. Ultimately, non-noble metallic electrocatalysts represent potential candidates for those made of noble metals, even though they are common on Earth. Non-metallic elements, including B, C, N, P, S and Se, can be used to construct HER electrocatalysts in composites with effective structures, including those with 0D, 1D, 2D and 3D structures [113–117].

**Table 1.** Summary of hydrogen evolution reaction (HER) properties of transition-metal-based catalysts.

| Catalyst | Electrolyte | BET Surface Area (m²/g) | Overpotential, η (mV) @ 10 mA/cm² | Tafel Slope, b (mV/dec) | Durability [1] |
|---|---|---|---|---|---|
| NiFe/Pt [68] | 0.1 M KOH | - | 128 | - | - |
| NiO/PtNi [70] | 1 M KOH | - | 39.8 | 78.8 | 10 h @ 10 mA/cm² |
| Ru-MoO$_2$ [75] | 1 M KOH and 0.5 M H$_2$SO$_4$ | - | 29 (1 M KOH), 55 (0.5 M H$_2$SO$_4$) | 31 (1 M KOH), 44 (0.5 M H$_2$SO$_4$) | |
| Ru-NMCNs-T [74] | 1 M KOH | 684 (Ru-NMCNs-T), 755 (NMCNs) | 28 | 57.6 | - |
| Ru-NiCCF [76] | 0.1 M NaOH | - | 100–300 | - | - |
| Ru@RuO$_2$ [77] | 0.1 M KOH | 39.8 | 137 | 112 | - |
| Ru@Co/N-CNTs [78] | 1 M KOH and 0.5 M H$_2$SO$_4$ | - | 48 (1 M KOH), 92 (0.5 M H$_2$SO$_4$) | 33 (1 M KOH), 45 (0.5 M H$_2$SO$_4$) | - |
| Ir$_{80}$Ni$_{20}$/Cu foam [83] | 1 M KOH | - | 60 | 40 | 10 h @ 10 mA/cm² |
| Ir NP/Siloxene [84] | | | | | - |
| Rh@Pt [85] | - | - | 139 | 84.8 | - |
| Rh@Ir [85] | - | - | 169 | 112 | - |
| Ir-Co [118] | 0.5 M H$_2$SO$_4$ | - | 23.9 | 25.7 | - |
| rGO-Fe$_3$O$_4$ [38] | 1 M KOH | | 310 | 80 | 24 h @ 10 mA/cm² |
| FeO$_x$-NBs [119] | 0.5 M KOH | | 450 | 85 | - |
| Co$_{0.8}$Ni$_{0.1}$Fe$_{0.1}$S$_2$ [120] | 0.5 M H$_2$SO$_4$ | | 138 | 49 | 33 h @10 mA/cm² |
| ZCS@rGO [121] | - | | 135 | 47 | 36 h @10 mA/cm² |
| FeMoO$_2$/MoO$_3$/ENF [122] | - | | 36 | - | 95 h @100 mA/cm² |
| CoCr$_2$O$_4$ [123] | 1 M KOH | | 212 | - | 24 h @10 mA/cm² |
| Fe2P@rGO [124] | - | | 101 | 55.2 | 12 h@10 mA/cm² |
| NiSx/NF [125] | 1 M KOH | | 53 | - | 100 h @100 mA/cm² |
| NiP/NiFeP/C [126] | 1 M KOH | | 1.53 V (cell) | - | 20 h @100 mA/cm² |
| Co-Fe-S@PB [127] | - | | 286 | 37.8 | 33 h @10 mA/cm² |
| CoZnCdCuMnS@CF [128] | 1 M KOH | | 173 | - | 70 h @10 mA/cm² |
| Au-MoS$_2$/CNFs [129] | - | | 92 | 126 | 50 h @10 mA/cm² |
| Ni3S$_2$ [130] | 1 M KOH | | 37.7 | - | 24 h @10 mA/cm² |
| NiFeSe/CFP [131] | 1 M KOH | | 186 | 52 | - |
| CozNiySx@PPy/CFP-6 (A-6) [132] | 1 M KOH | | 185 | 78 | 100 h @10 mA/cm² |
| Co$_7$Se$_8$ [133] | 1 M KOH | | 260 | 32.6 | 12 h @10 mA/cm² |
| MoS$_2$ [134] | - | | 190 | 54 | |
| NiCo$_2$S$_4$ nanosheet [135] | 1 M KOH | | 150 | 82 | - |
| (Ni-Fe)S-x/NiFe(OH)(y) [136] | 1 M KOH | | 1.46 V (cell) | - | 50 h @100 mA/cm² |
| NiP/NiFeP/C [126] | 1 M KOH | | 1.53 V (cell) | - | 20 h @100 mA/cm² |
| Co$_{0.75}$Ni$_{0.125}$Mn$_{0.125}$P [137] | - | | 137 | 49 | 12.5 @10 mA/cm² |
| CoWP-CA/KB [138] | - | | 111 | 58 | 60 h @10 mA/cm² |
| Multi-metal phosphide [139] | 0.5 M H$_2$SO$_4$ | | 220 | - | 100 h @100 mA/cm² |
| Cobalt phosphide nanoparticles [140] | 0.5 M H$_2$SO$_4$ | | 98 | 74 | - |
| MnRuPOGO-500 [141] | 1 M KOH | | 109 | 38.55 | 60 h @ 10 mA/cm² |
| Co-CoxC [142] | - | | 78 | 87 | 1 h @10 mA/cm² |
| Co-N-C [143] | 0.5 M H$_2$SO$_4$ | | 145 | - | - |
| Mo$_2$C-MoP hybrid nanodots [144] | 1 M KOH | | 147 | 64 | 120 @ 10 mA/cm² |
| N-Mo$_2$C/PC [145] | 0.5 M H$_2$SO$_4$ | | 109 | - | - |
| CoP$_3$/CoMoP/TiO$_2$-x@Ti [146] | - | | 143 | 61 | 48 h @ 10 mA/cm² |
| NiMo$_2$C@C [147] | 1 M KOH | | 181 | - | 10 h @ 10 mA/cm² |

[1] The following durability refers to chronopotentiometric (CP) tests.

## 4. MXenes as Emerging Materials for HER

MXene is a 2D nanomaterial based on transition-metal carbide or nitride, having the general formula of $M_{n+1}X_nT_x$, where M = transition metal, X = C and/or N and $T_x$ = surface termination groups such as F, O, OH and Cl. The n number varies from 1 to 4 [148–150]. MXene is fabricated from the etching of MAX phases, where their general formula is $M_{n+1}AX_n$. 'A' is the group 13–16 elements (i.e., Al, Ga), where n varies from 1 to 3. During etching, the 'A' layer is removed as the metallic bonding of the M-A bonds is weaker than the ionic/covalently bonded M-X bonds [149,151]. MXene has been extensively studied for the application of electrocatalytic HER as well as OER for water splitting. Past reviews have highlighted the clear potential of different types of MXenes based on Ti, Mo and V. Table 2 summarizes the HER properties of several common MXenes studied for

HER. Yet, the Ti-based MXenes, particularly the $Ti_3C_2T_x$ or $Ti_3C_2$, showed the majority. Termination groups are crucial for MXene's role in HER and as support. This is owing to their characteristics, including large surface area, metallic properties, high electron conductivity and the presence of hydrophilic termination groups [21]. Gibbs free energy for hydrogen adsorption ($\Delta G_{Hads}$) and intrinsic HER activity highly depend on 'M' transition metal and the surface termination groups. For instance, Mo-based $Mo_2CT_x$ MXenes are more catalytically active for HER than the most commonly used Ti-based MXenes [152]. O-termination groups also benefit HER. It has been shown that O-groups facilitate the desorption of H from the MXene surface, bringing the $\Delta G_{Hads}$ closer to optimum (zero). HER activity is limited if the F-termination coverage is high [153,154]. In terms of their durability, pristine MXenes such as $Ti_3C_2T_x$ are prone to oxidation within a short term (~12 days) when exposed to oxygen in the water. Modifications such as doping will potentially minimize oxidation to extend the MXene's durability [155].

**Table 2.** Summary of HER properties of commonly used MXenes under different structures and conditions.

| MXene | Structure | Electrolyte | Overpotential (mV) @ 10 mA/cm$^2$ | Tafel Slope (mV/dec) |
|---|---|---|---|---|
| $Ti_3C_2T_x$ [156,157] | Multilayer | 1 M KOH | >600 | - |
| $Ti_3C_2T_x$ [158] | Few layer | 1 M KOH | >500 | >100 |
| $Ti_3C_2T_x$ [152] | Few-layer | 0.5 M $H_2SO_4$ | 609 | 124 |
| $Ti_3C_2O_x$ [159] | Few-layer | 0.5 M $H_2SO_4$ | 190 | 60.7 |
| $Ti_3C_2(OH)_x$ [159] | Few-layer | 0.5 M $H_2SO_4$ | 217 | 88.5 |
| $Mo_2CT_x$ [152] | Few-layer | 0.5 M $H_2SO_4$ | 283 | 82 |
| $Mo_2CT_x$ [160] | Few-layer | 1 M KOH | 300 | 110 |
| $Mo_2CT_x$ [161] | Multilayer | 1 M KOH | 280 | 118 |

The electronic structure of MXene plays a role in the intrinsic activity toward HER. Electronic properties are affected by a number of factors, including the 'M' element, surface terminations, layer thickness, effects of intercalation, and adding dopants. Pristine $M_2X$ MXenes are primarily metallic. The presence of termination groups on the basal planes results in additional energy bands below the Fermi level that shift the MXene into a semiconductor, such as those of $Ti_2CO_2$. $Ti_3C_2T_x$ can exhibit metallic properties where it was found that F-groups occupy the face-centred cubic adsorption site while O-groups have a partial occupation on the bridge sites and hexagonal close-packed sites [162]. Further, electronic properties may vary between multilayer and few-layer $Ti_3C_2T_x$ given that few-layers may offer a larger in-plane conductivity [163,164]. Ti- and V-based MXenes also offer very high electron conductivity exceeding 1000 S/cm [165]. High electron conductivity is desired for active HER. The electronic properties are adjusted by doping the MXene and introducing the HER-active materials. MXene also affects the electronic properties of the interacting material. Kong et al. [166] found that the Ti site favours H adsorption in $Ti_3C_2O_2$ quantum dots (QDs). Graphene is able to form an interfacial interaction with the $Ti_3C_2O_2$ QDs that stabilizes the C—O configuration and shifts the d-band centre energy level by 0.4–0.5 eV in the $Ti_3C_2O_2$. The graphene-$Ti_3C_2O_2$ QDs with Gibbs free energy closer to zero value are more favourable towards HER. On the other hand, Ren et al. [167] reported that H-adsorption occurs on the O-sites within the hybrid of $MoS_2@Mo_2CO_2$. The catalyst also exhibits metallic properties and Gibbs free energy for hydrogen adsorption $\Delta G_{Hads}$ closer to optimal. Interaction between the $MoS_2$ and the $Mo_2CO_2$ is also interfacial through charge transfer. The exchange of electrons between the two components is one of the drivers of improved HER. Single-atom catalysts (SACs) and doping on MXene have positive outcomes for HER. In the case of Ru SACs on $Ti_3C_2T_x$ with N-doping, Ti—N and N—Ti—O bonds are formed after N-doping on the $Ti_3C_2T_x$. Ru SACs are attached in the form of pyrollic-N—Ru bonds. Interactions between MXene, Ru SACs and the N-groups result in a greater total density of state (TDOS) value indicating better electron conductivity. The partial density of state (PDOS) of $Ru_{SA}$-N-$Ti_3C_2T_x$ near the Fermi level is attributed

to the d-orbitals of Ti and Ru, where the Ru SACs brought about d-electron domination near the Fermi level that in turn benefits HER [168]. For transition-metal SACs, Co SACs in $V_2CT_x$ MXenes showed that electron transfer occurred between the Co SACs and $V_2CT_x$ through —O— bonds that facilitate early-stage water dissociation. The d-band centre of Co@$V_2CT_x$ is brought to an intermediate level and has high electron cloud distribution. d-band centre that is nearer the Fermi level is more favourable for adsorption/desorption of intermediates for both HER and OER [169]. Therefore, the electronic structure of MXene would tailor the electron conductivity and intrinsic activity towards HER. Dopants and introducing HER-active materials potentially result in electron redistribution that may lead to more favourable HER binding energies.

Another factor is that the morphology of MXenes can be tailored into several different morphological structures. For example, Multilayer MXenes are intercalated and then delaminated to form nanosheets of few-layer MXenes. Further modifications can produce structures with multiple pores [170] and unique structures such as crumpled [171], rolled [110], and spheres [172]. Various morphology changes the overall surface area of MXene, the termination groups/active sites' accessibility through the electrocatalytic active surface area (ECSA), electron conductivity, its ability to anchor the HER-active materials (Ni-based HER catalysts, single-atom catalysts, etc.) and their durability. This affects the overall HER activity of the MXene and MXene composites/hybrid catalysts. Several of these different morphologies of MXenes have been studied for HER.

## 5. HER Properties of Various MXene Structures

### 5.1. Multilayer and Few/Single-Layer MXenes

Multilayer MXenes are typically obtained after etching and exfoliating their MAX phase, particularly if the etchant is HF. Interlayer spacing increases after the removal of the 'A' layer (ex: Al), producing the morphology of multi-layered MXene displaying an accordion-like structure (an example is shown in Figure 5b). The HER properties can be influenced by the number of layers, thickness and interlayer spacing. Ultrathin layers with large spacing increase the surface area and access to active sites [24,173,174]. The packed structure of multilayer MXene stacks may limit the mass and ion transport.

On the other hand, few- or single-layer MXenes (sometimes called delaminated MXenes) are observed in the form of nanosheets rather than the accordion structure. These MXenes share similar roles as supporting material for HER electrocatalysts, whether multilayers or few layers. Types of MXenes and different termination groups significantly influence the intrinsic HER properties of MXenes. At the same time, the change in morphology determines the electrochemically active surface area and, therefore, the accessibility to the active sites. Li et al. [174] found that metallic ultrathin $Ti_3CT_x$ nanosheets have high electron conductivity and more exposed active areas than their bulk multi-layered counterpart, benefiting the HER by lowering the onset potential by as much as 190 mV. Furthermore, few-layer MXene nanosheets expressed larger electrochemical surface area (ECSA) with more exposed active sites than the multilayers [20]. Seh et al. [152] determined that delaminated $Mo_2CT_x$, an HER-active Mo-based MXene, possesses larger ECSA than the multilayer $Mo_2CT_x$. Basal planes of delaminated $Mo_2CT_x$ are more exposed, easing the access toward the termination groups. Exposing these termination groups can facilitate the anchoring of HER-active single atoms or nanoparticles/nanosheets. Furthermore, the electronic properties also vary with morphology. Regarding charge transfer resistance, few-layer $Ti_3C_2T_x$ can exhibit low resistance of 7.2 Ω [158] in 0.5 M $H_2SO_4$, while the multi-layered structure may have resistances as high as 200 Ω [155]. Faster electron transfer in a few layers may be due to its less-packed structure easing the electron transfer pathway. $Mo_2CT_x$ multilayer MXene ranges from 150 to 460 Ω in 1 M KOH, while the value for few-layer $Mo_2CT_x$ is unclear [161,167]. However, Liang et al. [161] determined that the charge transfer resistance of multilayer $Mo_2CT_x$ (461 Ω) is lower than $MoS_2$ (564 Ω), indicating better electron-conducting properties of the $Mo_2CT_x$ than the transition-metal $MoS_2$. The Mo-based MXene application in HER is still in the early stages with inconsistent

data. Hence, more studies are needed to understand the HER characteristics for Mo-based MXenes in different electrolytes.

Properties of multilayer MXenes may be altered by increasing the interlayer spacing, doping with heteroatoms, or simply introducing HER-active materials. Han et al. [175] showed the increase in this interlayer spacing through partial intercalation of $Ti_3C_2T_x$ while maintaining the accordion structure, allowing the N-doping to be carried out. N-functional groups are more effective HER-active sites that can improve HER activity in 0.5 M $H_2SO_4$. Huang et al. [176] synthesized $MoS_2$ nanosheets supported on $Ti_3C_2$. Figure 5 shows the SEM images of the accordion-like multilayer $Ti_3C_2$ and the heterostructure of $MoS_2/Ti_3C_2$. The $MoS_2$ enhanced the HER of pristine MXene, attributed to higher ECSA and fast electron transfer. Liang et al. [161] reported a 280 mV overpotential for multilayer $Mo_2CT_x$ in 1 M KOH, which further improved to 112 mV upon adding Co-$MoS_2$. MXenes effectively enhance the distribution of HER-active materials. Recently, Wu et al. [177] synthesized multi-layered S-doped $Nb_4C_2T_x$. $NbS_2$ nanoparticles were observed upon S-doping, where they occupied between the layers of the MXene's accordion-like structure. High electron conductivity, good distribution of $NbS_2$ and large interlayer spacing with available pores contribute to an overpotential of 118 mV at 10 mA/cm$^2$ in 1 M KOH, while the overpotential of pristine $Nb_4C_2T_x$ was 324 mV. This study reported the potential of using other types of MXene besides $Ti_3C_2T_x$ or $Mo_2CT_x$ in HER application.

The issue facing few-layer MXenes is their tendency to restack and agglomerate due to strong van der Waals forces, rendering the active sites unavailable to participate in HER or anchoring HER-active phases. Cui et al. [178] showed that the Pt nanoparticles attached to few-layer $Ti_3C_2T_x$ could prevent the nanosheets from restacking. Further, combining $Ti_3C_2T_x$ and Pt with single-walled carbon nanotube (SWCNT) yielded a good HER electrocatalyst having 62 mV overpotential at 10 mA/cm$^2$ in 0.5 M $H_2SO_4$. Furthermore, a chronopotentiometric test revealed excellent durability of the hybrid catalyst throughout 800 h at constant 10 mA/cm$^2$. In another study, Yue et al. [179] prepared vertically aligned NiCo nanosheets on a few-layer $Ti_3C_2$ MXene that also hindered the MXene from restacking through spatial hindrance. Further, Du et al. [180] carried out Nb-doping on few-layer $Ti_3C_2T_x$ to enhance its electrical conductivity, followed by incorporating a NiCo alloy as the HER-active material. Despite the ECSA appearing to be reduced, the overpotential of optimized NiCo supported on Nb-doped $Ti_3C_2T_x$ was 43.4 mV at 10 mA/cm$^2$ in 1 M KOH, comparable to 10% Pt/C at 34.4 mV. The HER performance as well as durability of the catalyst, denoted as NiCo@NTM, is shown in Figure 6.

*5.2. Porous MXenes*

The morphologies of porous Mxenes are those with well-defined macro/micro/ mesoporous structures. Introducing the pores onto the MXene can be carried out in different approaches, yielding various porous MXenes. The availability of pores allows more favourable access of electrolyte materials towards the active sites, thus enlarging the electrochemically active surface area. Using porous scaffolds such as nickel foam (NF) is one direct method to introduce the porous morphology in an electrode material. Few-layer MXenes may be coated on the surface of NF to function as additional supports for electrocatalyst nanoparticles/nanosheets and boost their electron conductivity as well as homogeneity. Yu et al. [181] have reported that plane-to-plane coverage of MXene nanosheets on macroporous NF helps assist the even growth of NiFe-layered double hydroxide (LDH). MXene makes the NF more hydrophilic, thus allowing the electrolyte to permeate through the pores of NF and mesopores of NiFe-LDH. The MXene-coated scaffold improves the otherwise limited HER activity of NiFe-LDH. The HER in 1 M KOH of NiFe-LDH/MXene/NF was less than that of Pt/C/NF at low current density but exceeded at higher current density. The required overpotential was 205 mV at 500 mA/cm$^2$, whereas Pt/C/NF was 366 mV. The better HER was attributed to the Volmer step's acceleration and porous channels' existence. In addition, the NiFe-LDH/MXene/NF displays excellent OER activity, given

the benefit of NiFe-based catalysts for OER. Durability at 280 h was also improved due to the binder-free characteristics that have stronger adhesion of the nanosheets on MXene/NF.

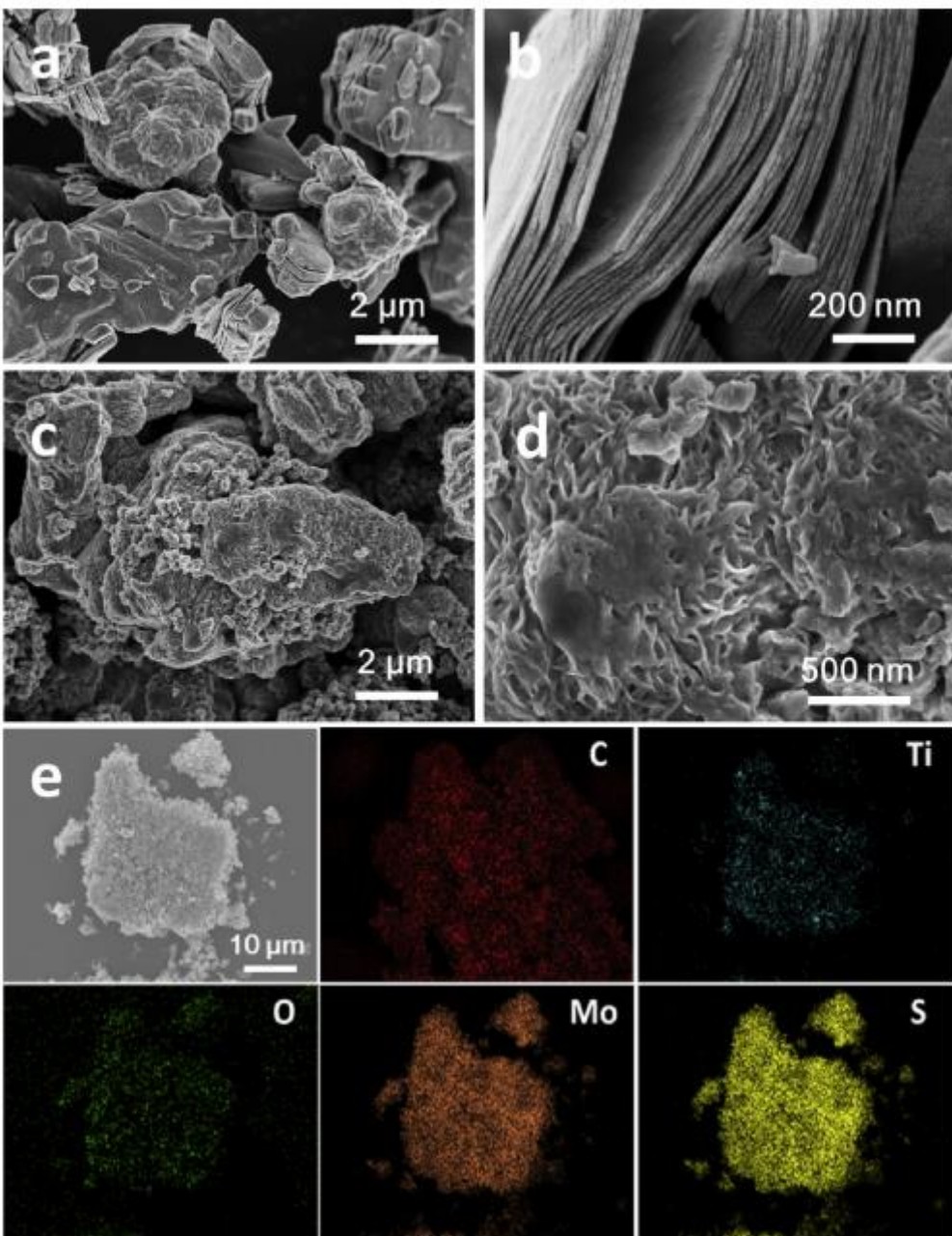

**Figure 5.** SEM images of Ti$_3$C$_2$ nanosheets (**a**,**b**) and MoS$_2$/Ti$_3$C$_2$ heterostructures (**c**,**d**). Elemental mapping images (**e**) of MoS$_2$/Ti$_3$C$_2$ heterostructures. Adapted with permission from [176], Copyright 2019, Elsevier.

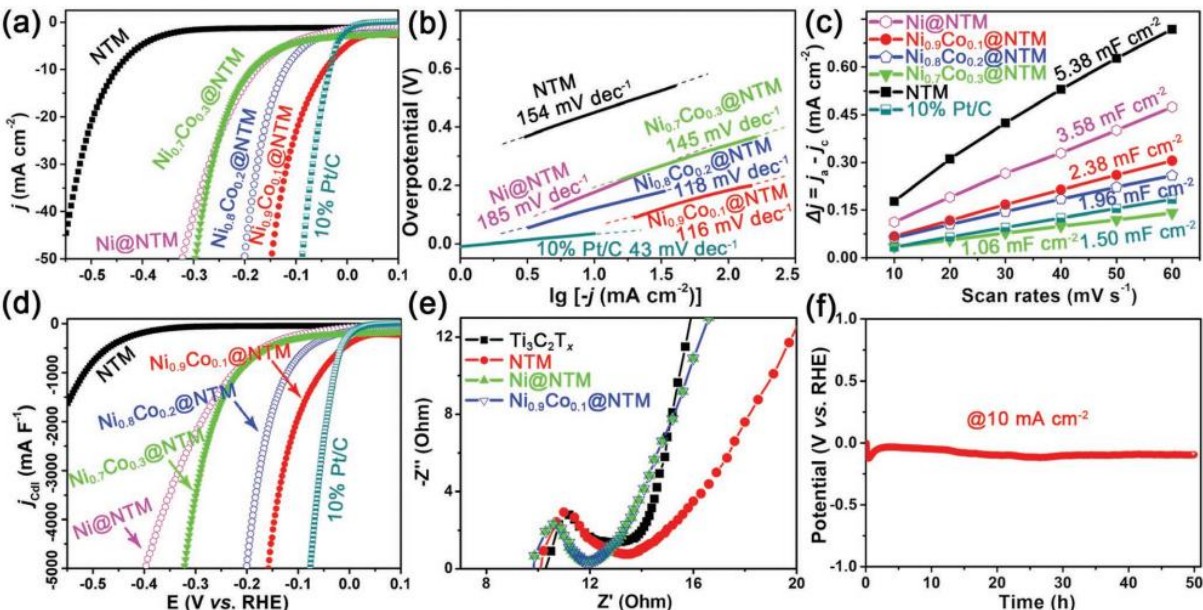

**Figure 6.** (**a**) HER polarization curves of the series NiCo@NTM nanohybrids, Ni@NTM nanohybrid, NTM and 10% Pt/C in 1M KOH with a scan rate of 5 mV/s. (**b**) Corresponding Tafel plots of the series samples. (**c**) The charging current density differences plotted against scan rates of the as-prepared series NiCo@NTM and Ni@NTM nanohybrids. The linear slope is equivalent to twice the electrochemical double-layer capacitance (Cdl). (**d**) HER polarization curves shown in (**a**) normalized by the Cdl. (**e**) Nyquist plots of the electrodes modified by $Ti_3C_2T_x$, NTM, Ni@NTM, and $Ni_{0.9}Co_{0.1}$@ NTM measured at zero overpotential versus RHE. (**f**) The chronopotentiometric curves of the $Ni_{0.9}Co_{0.1}$@NTM nanohybrid under static current density (10 mA/cm$^2$) over 50 h. Adapted from [180].

When coating MXene on porous scaffolds, proper adsorption of MXene onto these scaffolds must be considered to ensure the close interaction between the two materials for fast electron transfer. Therefore, optimizing the coating conditions during preparation should be taken into account. On the other hand, it can be beneficial to introduce those porous channels on the MXene itself, which has been shown to not only enlarge the surface area but also expose more edge and defect sites, prevent the restacking of MXene nanosheets and effectively anchor the electrocatalytically active materials. Among the types of porous MXenes that have been successfully synthesized are the in-plane porous (holey), MXene foams and reassembly of MXenes into a three-dimensional (3D) porous structure [61,170]. Yet, it is still important to note that the porous MXenes have very limited intrinsic HER activity. Therefore, these MXenes can still be utilized as supports for HER-active materials.

In-plane porous (holey) MXenes display multiple nanosized holes on their basal planes and edges. Le et al. [182] utilized chemical etching to create holes on the planes and edges of $Ti_3C_2T_x$, which are then used as support for IrCo nanoparticles. These holes attribute to the meso/micro/macroporous structure on the $Ti_3C_2T_x$ (denoted as ac-$Ti_3C_2T_x$), which greatly improves BET surface area from 6.5 m$^2$/g (pristine multilayer $Ti_3C_2T_x$) and 19.6 m$^2$/g (delaminated $Ti_3C_2T_x$) to around 189.1 m$^2$/g (porous ac-$Ti_3C_2T_x$) and 175 m$^2$/g (for IrCo/ac-$Ti_3C_2T_x$). The slight reduction of surface area may be due to the occupancy of IrCo on the pores of MXene. HER overpotential of IrCo/ac-$Ti_3C_2T_x$ in 1 M KOH was around 135 mV for 10 mA/cm$^2$. In addition, the catalyst was also tested for OER and revealed a 220 mV overpotential at the same current density, thus displaying bifunctional properties. Furthermore, 98% of the HER current density of IrCo/ac-$Ti_3C_2T_x$ was maintained after 30 h. Kang et al. [183] adopted a similar chemical etching approach to introduce nanometre-sized pores (5–10 nm) on basal planes of $Ti_3C_2T_x$ nanosheets. Compared to pristine $Ti_3C_2T_x$, surface potential of the porous MXene is less negative due to the removal

of surface functional groups when pores were created. While some important surface functional groups are lost, it is still sufficient to adhere and disperse the NiCoP nanoparticles uniformly. BET surface area of the $Ti_3C_2T_x$ improves by 21.91 $m^2/g$ upon introducing porous structure and increases by 49.12 $m^2/g$ upon the addition of NiCoP. The resulting HER overpotential was 101 mV (1 M KOH) and 115 mV (0.5 M $H_2SO_4$) at 10 $mA/cm^2$, while pristine $Ti_3C_2T_x$@NiCoP shows 121 mV (1 M KOH) and 134 mV (0.5 M $H_2SO_4$); thus, improvement was contributed by the porous structure towards HER. The HER performance in 0.5 M $H_2SO_4$ is shown in Figure 7a,b, as well as its durability in Figure 7f. Recently, Kong et al. [184] synthesized $Pt_3Ni$ nanoparticles supported on $Ti_3C_2T_x$ through solvothermal method, followed by etching in the presence of polystyrene spheres to create the in-plane porous structure of hybrid $Pt_3Ni$-$Ti_3C_2T_x$, giving an inverse-opal structure. The porous $Pt_3Ni$-$Ti_3C_2T_x$ improved the HER to some extent, with the overpotential of 44.1 mV at 10 $mA/cm^2$ in 1 M KOH, compared to its non-porous counterpart at 46.8 mV. Indeed, in-plane pores on MXene ease the flow of electrolyte materials and products between active sites and surroundings, which benefits HER. Loss of termination groups and gaps due to the holes may impede the intrinsic HER of the MXene along with its electron conductivity, which can be compensated by including the HER-active nanoparticles/nanosheets. However, it is crucial to control the size of the holes/pores to maintain high electron conductivity and surface area to anchor the active phases.

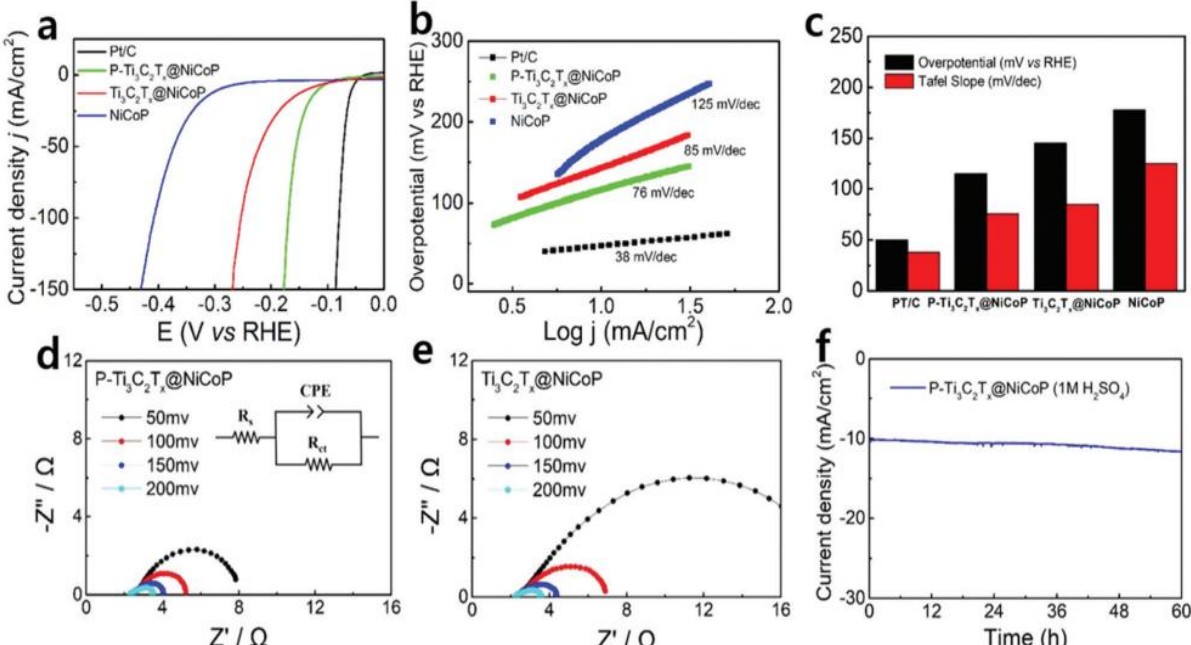

**Figure 7.** Electrocatalytic HER performances of various samples in acid. (**a**) HER polarization curves (95% iR-compensated) measured at 5.0 mV/s in 0.5 M $H_2SO_4$, (**b**) corresponding Tafel plots, (**c**) overpotential, Tafel summary plot at 5.0 mV/ s in 0.5 M $H_2SO_4$, Nyquist plots of (**d**) P-$Ti_3C_2Tx$@NiCoP, (**e**) $Ti_3C_2Tx$@NiCoP measured at different scan rates in 0.5 M $H_2SO_4$ and (**f**) current density-time (I-t) curves at an overpotential of 125 mV. Adapted with permission from [183], Copyright 2021, Royal Society of Chemistry.

Reassembly of MXene nanosheets can create structures containing multiple porous channels to facilitate electrolyte/product passage rather than creating holes. Recently, Lin et al. [185] reported the self-assembly of $Ti_3C_2T_x$ nanosheets into a 3D porous network through the assistance of $H^+$ melamine. –N and –S heteroatoms incorporated into the porous channels are able to stabilize the Ir single-atom catalysts (SACs) to create the active phase for HER. As a result, BET surface area improves significantly from 24.795 $m^2/g$ for pristine $Ti_3C_2T_x$ nanosheets to 107 $m^2/g$ for Ir SA-2NS-$Ti_3C_2T_x$. The overpotential of the Ir

SA-2NS-Ti$_3$C$_2$T$_x$ network was 57.7 mV and 40.9 mV at 10 mA/cm$^2$ in 0.5 M H$_2$SO$_4$ and 1 M KOH, respectively. The mass activity was at around 309.6 mA/mg$_{Ir}$ in acid, much larger than Pt/C (22.9 mA/mg$_{Pt}$). Furthermore, Peng et al. [186] found that the crosslinking between Ti$_3$C$_2$T$_x$ nanosheets to form the 3D porous network can also prevent the MXene sheets from spontaneously aggregating. Similar to the Ir SACs, supporting Pt SACs on the porous Ti$_3$C$_2$T$_x$ networks yields an active HER in both acid and alkaline conditions. The overpotential of Pt SAC on porous Ti$_3$C$_2$T$_x$ is as low as 35 mV at 10 mA/cm$^2$ in 0.5 M H$_2$SO$_4$ and appeared stable after 60 h chronoamperometry tests.

Other 2D nanomaterials, such as graphene and graphitic carbon nitride (g-C$_3$N$_4$), can be incorporated together with MXene nanosheets, constructing a multicomponent 3D porous structure. Three-dimensional porous N-doped graphene (NG)/Ti$_3$C$_2$T$_x$ networks comprise macropores of nanometer and micrometer sizes that help expose active sites, as shown by Shen et al. [187]. NG/Ti$_3$C$_2$T$_x$ has higher electron conductivity than individual Ti$_3$C$_2$T$_x$, as well as a large BET surface area of 148.2 m$^2$/g. N-doping further tunes the graphene's electronic structure to make the hybrid structure more favorable to HER. However, a high overpotential of 354 mV at 10 mA/cm$^2$ was observed in acid conditions, which may be due to insufficient active sites provided by Ti$_3$C$_2$T$_x$. Therefore, the composition of NG and MXene should be carefully optimized to ensure sufficient electron conductivity and availability of active sites on the porous structure. Recently, He et al. [158] improved the properties of the multicomponent structure by combining Ti$_3$C$_2$T$_x$ nanosheets, RGO, and g-C$_3$N$_4$ into one self-assembled 3D porous network (MX/CN/RGO). The structure revealed a high BET surface area of 345.6 m$^2$/g attributed to the wide range of nanopores, and the restacking/aggregation of individual nanosheets is avoided. The pores can be clearly seen in the FESEM image shown in Figure 8a,b. Furthermore, low charge transfer resistance and abundant active sites provided by N-groups in g-C$_3$N$_4$ alongside O-termination on Ti$_3$C$_2$T+ bring about more optimized H-adsorption Gibbs free energy. HER-onset overpotential for MX/CN/RGO was at 38 mV and 382 mV, which is required to reach 100 mA/cm$^2$. Stability of the porous catalyst was acceptable after around 5 h of chronoamperometry testing and 2000 CV cycles.

These porous MXenes utilized for HER electrocatalysts show similar porous characteristics that facilitate the passage of electrolyte material and ions to reach the active sites and the movement of products away from the catalyst. They clearly exhibited larger surface areas than pristine multilayer and few-layer MXenes. Charge transfer resistance may increase, and loss of active sites can occur for in-plane porous MXenes, but they still effectively adhere to the HER-active materials for better dispersion. Doping with heteroatoms may improve HER of in-plane porous MXenes. Self-assembled MXene nanosheets into porous networks and their hybrid multicomponent with graphene and g-C$_3$N$_4$ also showed potential HER activity. Further optimization is necessary to balance the properties, in particular electron conductivity, the concentration of active sites and the strength of the 3D porous network to ensure the electrocatalyst is stable in acidic as well as alkaline conditions. The binary/ternary 3D porous network can also be further adopted as support for nanoparticles, nanosheets and SACs. Furthermore, other types of MXenes besides Ti$_3$C$_2$T$_x$ in their porous forms should also be pursued to understand their feasibility for HER.

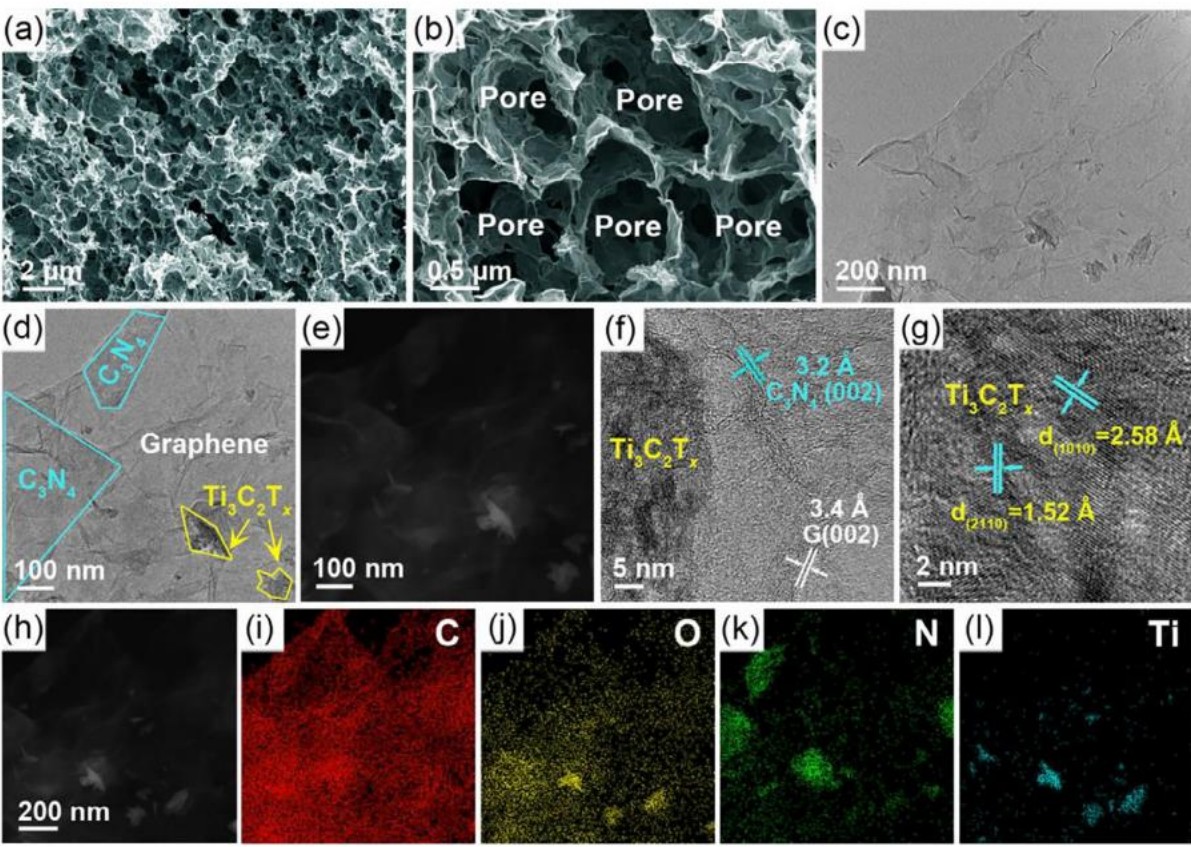

**Figure 8.** Morphological and microstructural analysis of the 3D MX/CN/RGO nanoarchitecture. Representative (**a**,**b**) FESEM, (**c**,**d**), TEM and (**e**) HAADF-STEM images reveal the successful integration of $Ti_3C_2T_x$, g-$C_3N_4$ nanosheets and graphene into a 3D interconnected framework. (**f**,**g**) HR-TEM images disclose the lattice fringes of $Ti_3C_2T_x$ and g-$C_3N_4$ nanosheets. (**h**) HAADF-STEM image and corresponding elemental mapping analysis demonstrate that the 3D hybrid nanoarchitecture is composed of (**i**) C, (**j**) O, (**k**) N and (**l**) Ti elements. Adapted with permission from [158], Copyright 2022, Elsevier.

### 5.3. Special Structures—Crumple and Rolled MXenes

MXenes with special structures can be synthesized following additional steps after obtaining delaminated MXenes. Two known interesting morphologies are crumpled and rolled MXenes. To date, limited reports are available on studying crumpled and rolled MXene applications for electrocatalytic HER. The two examples mentioned showed the potential of these MXenes in HER. Wu et al. [171] prepared $Ti_3C_2T_x$ with crumpled paper-like morphology utilized as support for nano-sized Pt clusters. The crumpled $Ti_3C_2T_x$, prepared from spray drying of the delaminated MXene, showed the 3D crumpled paper ball morphology and did not change in appearance when Pt was introduced, as shown in Figure 9b–d. Benefiting from the enlarged ECSA, four-times-larger BET surface area than freeze-dried $Ti_3C_2T_x$, thin layer, less restacking and stabilization of Pt clusters by Ti-O termination, Pt-supported crumpled $Ti_3C_2T_x$ with 2.9 wt.% Pt loading has an overpotential of 34 mV at 10 mA/cm$^2$, on par with Pt/C at 37 mV in 0.5 M $H_2SO_4$. The crumpled $Ti_3C_2T_x$ itself has a limited HER activity with 247 mV overpotential; thus, introducing an active material is still necessary. Mass activity Pt-supported crumpled $Ti_3C_2T_x$ was also found to be seven times higher than Pt/C, in addition to its good stability within the 10,000 s durability test. Aside from crumpled morphology, Liu et al. [110] synthesized $Ti_3C_2T_x$ with an interesting nanoroll morphology. The morphology of the rolled MXenes appears as thin rolled papers with open ends due to rapid freeze-drying and shrinking of MXene nanosheets. The addition of $MoS_2$ and annealing does not alter the structure,

where the $MoS_2/Ti_3C_2T_x$ nanorolls exhibit ~200 nm diameter and ~20 μm length. The hybrid $MoS_2/Ti_3C_2T_x$ nanorolls showed increased charge and mass transfer with good dispersion of vertically aligned $MoS_2$. HER overpotential in 0.5 M $H_2SO_4$ was 70 mV at 10 mA/cm$^2$, much lower than the individual components (101 mV for $MoS_2$ and 162 mV for $Ti_3C_2T_x$). Twelve-hour stability tests showed negligible change in HER and no change in the nanoroll morphology.

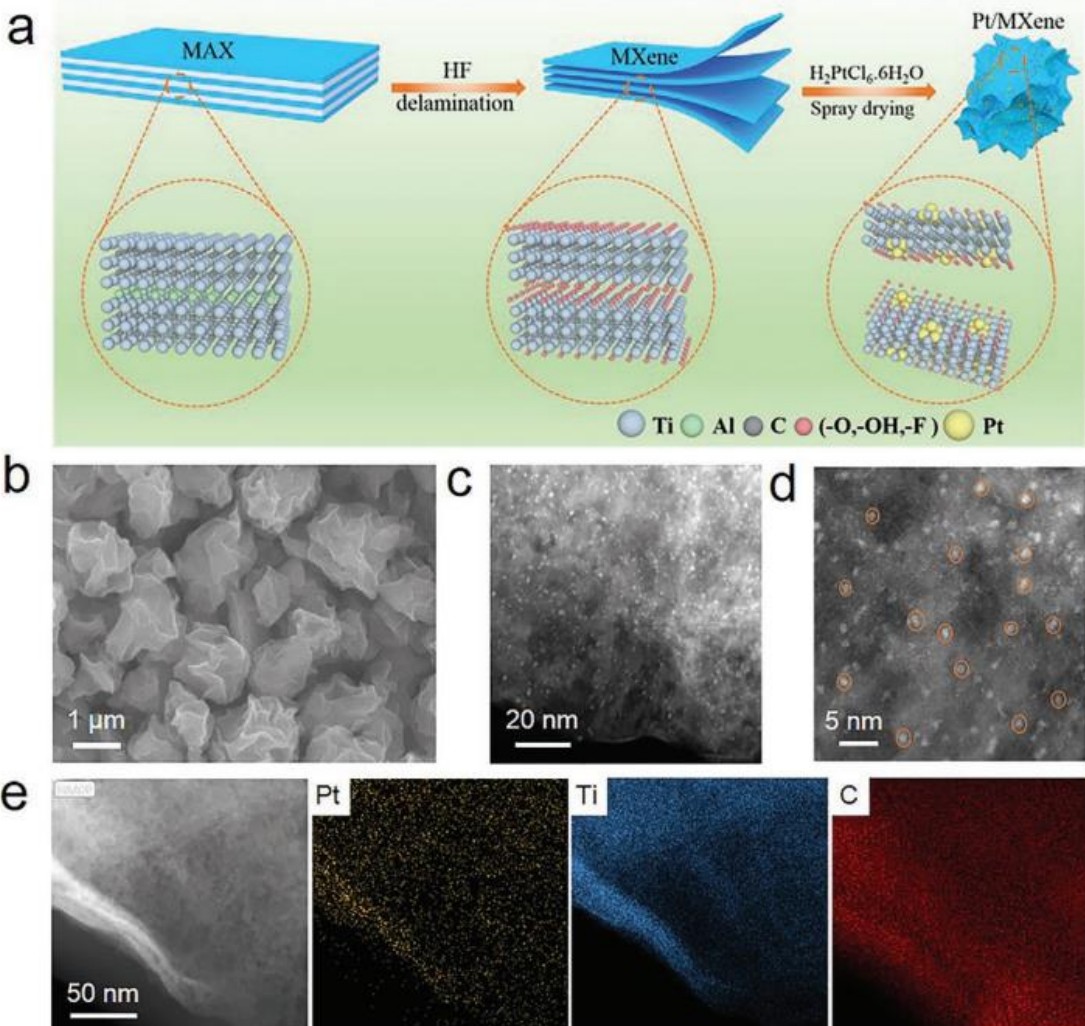

**Figure 9.** (**a**) Schematic illustration for the preparation of Pt/MXene. (**b**) SEM images of Pt/MXene. (**c**,**d**) HAADF-STEM images of Pt/MXene and (**e**) the EDX elemental mappings. Adapted with permission from [171], Copyright 2022, John Wiley and Sons.

The abovementioned examples of crumpled and rolled MXenes have been utilized as supports for Pt and $MoS_2$, where their HER activity was determined under acidic conditions. In both cases, the short-term durability was also reported to be stable. Other opportunities, such as the difference in ECSA and HER activity/durability of the MXene itself, should be explored compared with multilayer/few-layer MXene. Furthermore, there is also the potential to study these MXenes under alkaline conditions and support other HER-active materials such as NiCo and NiMo.

In summary, the change in morphology of MXenes is still able to maintain the roles of MXenes as effective supports for HER-active materials. Through this change, it is possible to minimize the aggregation and restacking for both MXenes and catalyst nanoparticles/nanosheets. Furthermore, the surface area is clearly enlarged, particularly for the MXenes with porous structures. It can be seen that the 3D porous frameworks such as 3D

MX/CN/RGO achieved a BET surface area of more than 300 m$^2$/g, much larger than its individual components. Table 3 summarizes the HER properties of MXene and MXene composites, which have different MXene structures. Several catalysts with MXene have comparable performance to that of catalysts without MXenes. Pt-containing catalysts with MXene supports are expected to show low overpotentials. Pt SAC on a porous Ti$_3$C$_2$T$_x$ framework has a comparable HER performance to that of NiO/PtNi, with the former having a slightly lower overpotential by roughly 4 mV. This can be attributed to the well-defined porosity of the Pt SAC-porous Ti$_3$C$_2$T$_x$ and homogenous dispersion of Pt SACs. A combination of RGO and transition metals such as Fe$_2$P shows improvement in HER activity owing to the high electron conductivity. RGO as well as g-C$_3$N$_4$ are compatible with MXene such as the properly synthesized 3D MXene/RGO/g-C$_3$N$_4$ exhibiting high BET surface area and excellent HER properties. These hybrid structures require further exploration for supporting HER-active nanoparticles such as NiCo and NiMo. Additionally, MoS$_2$ HER overpotential appeared lower when supported on rolled MXene than the multi-layer MXene, showing the effects of changing morphology. Other HER-active catalysts, as shown in Table 1 such as NiCo$_2$S$_4$, Ni$_3$S$_2$, and NiP may be integrated together with MXenes that may further boost their dispersion, HER properties and stability. Most catalysts exhibit acceptable durability at low current densities, even for the non-MXene-containing catalysts and the non-modified MXenes. Durability should be considered at longer durations (>300 h) at high current densities.

**Table 3.** Summary of hydrogen evolution reaction (HER) properties of MXene-based catalysts.

| MXenes | | | | | | | |
|---|---|---|---|---|---|---|---|
| Catalyst | MXene Type | MXene Morphology | Electrolyte | BET Surface Area (m$^2$/g) | Overpotential, η (mV) @ 10 mA/cm$^2$ | Tafel Slope, b (mV/dec) | Durability |
| Multilayer MXene | | | | | | | |
| Co-MoS$_2$/Mo$_2$CT$_x$ [161] | Mo$_2$CT$_x$ | Multilayer | 1 M KOH | - | 112 | 82 | LSV loss (1000 cycles): negligible. CA @ 10 mA/cm$^2$, 18 h: slight decline in current density. |
| S-ML-Nb$_4$C$_3$T$_x$ [177] | Nb$_4$C$_3$T$_x$ | Multilayer | 1 M KOH | 45.15 (S-ML-Nb$_4$C$_3$T$_x$) 31.33 (ML-Nb$_4$C$_3$T$_x$) | 118 | 104 | LSV loss (2000 cycles): slight increase in overpotential at 10 mA/cm$^2$. CA @ 10 mA/cm$^2$, 24 h: steady current density. |
| NiS$_2$/T-MXene [188] | Ti$_3$C$_2$T$_x$ | Multilayer | 1 M KOH | 28.1 (NiS$_2$/Ti-MXene) 4.7 (pristine multilayer Ti-MXene) | - | 100 | - |
| MoS$_2$/Ti$_3$C$_2$ [176] | Ti$_3$C$_2$ | Multilayer | 0.5 M H$_2$SO$_4$ | - | 280 | 68 | CP @ 10 mA/cm$^2$, 35 h: slight increase in potential within 22 h. Potential decrease until 35 h. |
| N-MXene-35 [175] | Ti$_3$C$_2$T$_x$ | Multilayer | 0.5 M H$_2$SO$_4$ | 23.6 (N-MXene-35) 12.3 (pristine multilayer Ti$_3$C$_2$T$_x$) | 162 | 69 | LSV loss (24 h CV cycling): slight increase in overpotential. CP @ 10 mA/cm$^2$, 35 h: steady potential. |
| LiF + HCl-etched Ti$_3$C$_2$T$_x$ [157] | Ti$_3$C$_2$T$_x$ | Multilayer | 0.5 M H$_2$SO$_4$ | - | 538 | 128 | - |
| Mo$_2$CT$_x$ [157] | Mo$_2$CT$_x$ | Multilayer | 0.5 M H$_2$SO$_4$ | - | 189 | 75 | LSV loss (1000 cycles): negligible |
| Ti$_2$CT$_x$ [152] | Ti$_2$CT$_x$ | Multilayer | 0.5 M H$_2$SO$_4$ | - | 609 | | CP @ 10 mA/cm$^2$, 120 h: steady potential. |

**Table 3.** *Cont.*

| Catalyst | MXene Type | MXene Morphology | Electrolyte | BET Surface Area (m²/g) | Overpotential, η (mV) @ 10 mA/cm² | Tafel Slope, b (mV/dec) | Durability |
|---|---|---|---|---|---|---|---|
| | | | | **MXenes** | | | |
| | | | | Few-layer MXene | | | |
| NiS$_2$/V-MXene [188] | V$_2$CT$_x$ | Few-layer | 1 M KOH | 44.4 (NiS$_2$/V-MXene) 7.5 (pristine V-MXene) | 179 | 85 | CP @ 10 mA/cm², 96 h: Slight potential increase. Post-CP LSV test also shows overpotential increase. |
| Ti$_3$C$_2$T$_x$@mNiCoP [179] | Ti$_3$C$_2$T$_x$ | Few-layer | 1 M KOH | 143.5 (Ti$_3$C$_2$@mNiCoP) | 127 | 103 | CA @ 10 mA/cm², 10 h: <5% current density loss. |
| NiCo@Nb-doped Ti$_3$C$_2$T$_x$ [180] | Ti$_3$C$_2$T$_x$ | Few-layer | 1 M KOH | - | 43.4 | 116 | CP @ 10 mA/cm², 50 h: steady potential |
| Ti$_3$C$_2$O$_x$ [159] | Ti$_3$C$_2$T$_x$ | Few-layer | 0.5 M H$_2$SO$_4$ | - | 190 | 60.7 | LSV loss (2000 cycles): slight overpotential increase. LSV loss (10,000 cycles): negligible. |
| Pt SA-Mo$_2$TiC$_2$T$_x$ [189] | Mo$_2$TiC$_2$T$_x$ | Few-layer | 0.5 M H$_2$SO$_4$ | - | 30 | 30 | CA @ 100 mA/cm², 100 h: slight decline in current density. |
| NiSe$_2$/Ti$_3$C$_2$T$_x$ [190] | Ti$_3$C$_2$T$_x$ | Few-layer | 0.5 M H$_2$SO$_4$ | - | 200 | 37.7 | LSV loss (2000 cycles): negligible. CA @ 10 mA/cm², 10 h: slight decline in current density. |
| S-M-5Pt [178] | Ti$_3$C$_2$T$_x$ | Few-layer | 0.5 M H$_2$SO$_4$ | - | 62 | 78 | CP @ 10 mA/cm², 800 h: slight potential loss. |
| | | | | Porous MXenes | | | |
| P-MoO$_2$ FCL/MXene/NF [191] | Ti$_3$C$_2$T$_x$ | Multilayer on porous scaffold (NF) | 1 M KOH | 5.96 (mesopores of P-MoO$_2$ FCL/MXene/NF) | 179 | 40.44 | CA @ 10 mA/cm², 42 h: 99.73% maintained current density. |
| NiFe-LDH/MXene/NF [181] | Ti$_3$C$_2$T$_x$ | Few-layer on porous scaffold (NF) | 1 M KOH | - | 132 | 70 | CP @ 10 mA/cm², 280 h: steady potential. |
| Pt$_3$Ni/Ti$_3$C$_2$T$_x$ [184] | Ti$_3$C$_2$T$_x$ | In-plane porous | 1 M KOH and 0.5 M H$_2$SO$_4$ | - | 46.8 (1 M KOH), 30 (0.5 M H$_2$SO$_4$) | 44.15 (1 M KOH), 26.51 (0.5 M H$_2$SO$_4$) | CA @ 10 mA/cm², 10 h: steady current density (1 M KOH and 0.5 M H$_2$SO$_4$). |
| IrCo@ac-Ti$_3$C$_2$ [182] | Ti$_3$C$_2$ | In-plane porous | 1 M KOH | 175 (IrCo@ac-Ti$_3$C$_2$) 189.1 (ac-Ti$_3$C$_2$) 6.5 (pristine multilayer Ti$_3$C$_2$) 19.6 (delaminated Ti$_3$C$_2$) | 135 | 56 | CA @ 10 mA/cm², 30 h: 98% maintained current density. |
| 3D MX/NG [187] | Ti$_3$C$_2$T$_x$ | MXene-graphene porous network | 0.5 M H$_2$SO$_4$ | 148.2 (MX/NG) 12.2 (pristine few-layer Ti$_3$C$_2$T$_x$) | 354 | 84 | LSV loss (2000 cycles): negligible. CA @ 10 mA/cm², 5000 s: slight decline in current density |
| 3D MX/CN/RGO [158] | Ti$_3$C$_2$T$_x$ | MXene-g-C$_3$N$_4$-RGO porous network | 0.5 M H$_2$SO$_4$ | 345.6 (3D MX/CN/RGO) 11.2 (GO) 4 (g-C$_3$N$_4$) 12.2 (pristine few-layer Ti$_3$C$_2$T$_x$) | 38 | 76 | LSV loss (2000 cycles): negligible. CA @ 10 mA/cm², 20,000 s: steady current density. |
| NiFe LDH/MX-rGO [192] | Ti$_3$C$_2$T$_x$ | MXene-rGO porous network | 0.5 M H$_2$SO$_4$ | 254.7 (NiFe LDH/MX-rGO) 116.4 (bare LDH) 12.2 (pristine few-layer Ti$_3$C$_2$T$_x$) | 326 | 100 | CA@ 20 mA/cm², 40 h: steady current density. |
| Pt-Porous Ti$_3$C$_2$T$_x$/Ti$_3$AlC$_2$ monolith [193] | Ti$_3$C$_2$T$_x$ | In-plane porous | 0.5 M H$_2$SO$_4$ | - | 37 | 89 | CP @ 10 mA/cm², 10 h: slight potential increase. |

**Table 3.** *Cont.*

| | | | | MXenes | | | |
|---|---|---|---|---|---|---|---|
| **Catalyst** | **MXene Type** | **MXene Morphology** | **Electrolyte** | **BET Surface Area (m$^2$/g)** | **Overpotential, η (mV) @ 10 mA/cm$^2$** | **Tafel Slope, b (mV/dec)** | **Durability** |
| P-Ti$_3$C$_2$T$_x$@NiCoP [183] | Ti$_3$C$_2$T$_x$ | In-plane porous | 1 M KOH and 0.5 M H$_2$SO$_4$ | 80.09 (P-Ti$_3$C$_2$T$_x$@NiCoP) 9.06 (pristine multilayer Ti$_3$C$_2$T$_x$) 30.97 (porous Ti$_3$C$_2$T$_x$) | 101 (1 M KOH), 115 (0.5 M H$_2$SO$_4$) | 69 (1 M KOH), 76 (0.5 M H$_2$SO$_4$) | CA @ 10 mA/cm$^2$, 1 M KOH and 0.5 M H$_2$SO$_4$, 60 h: slight decline in current density. |
| Ir$_{SA}$-2NS-Ti$_3$C$_2$T$_x$ [185] | Ti$_3$C$_2$T$_x$ | Self-assembled porous framework | 1 M KOH and 0.5 M H$_2$SO$_4$ | 107 (Ir$_{SA}$-2NS-Ti$_3$C$_2$T$_x$) 24.795 (pristine few-layer Ti$_3$C$_2$T$_x$) | 40.9 (1 M KOH), 57.7 (0.5 M H$_2$SO$_4$) | 50.5 (1 M KOH), 25 (0.5 M H$_2$SO$_4$) | LSV loss (10,000 cycles, 0.5 M H$_2$SO$_4$): slight increase overpotential @ 10 mA/cm$^2$. |
| Pt SA-PNPM [186] | Ti$_3$C$_2$T$_x$ | Self-assembled porous framework | 1 M KOH and 0.5 M H$_2$SO$_4$ | 121 (Pt SA-PNPM) 38.79 (freeze-dried obtained pristine Ti$_3$C$_2$T$_x$) | 36 (1 M KOH). 35 (0.5 M H$_2$SO$_4$) | 33 (1 M KOH), 31 (0.5 M H$_2$SO$_4$) | LSV loss (5000 cycles, 0.5 M H$_2$SO$_4$): negligible. CA @ 10 mA/cm$^2$, 60 h: negligible loss. |
| | | | | Special structures MXene | | | |
| Pt/Crumpled MXene [171] | Ti$_3$C$_2$T$_x$ | Crumpled | 0.5 M H$_2$SO$_4$ | 7.2 (crumpled spray-dried Ti$_3$C$_2$T$_x$) 1.9 (pristine freeze-dried Ti$_3$C$_2$T$_x$) | 34 | 29.7 | CP @ 10 mA/cm$^2$, 10,000 s: 9 mV potential drop. |
| MoS$_2$/Ti$_3$C$_2$T$_x$ nanoroll [110] | Ti$_3$C$_2$T$_x$ | Nanoroll | 0.5 M H$_2$SO$_4$ | - | 152 | 70 | LSV loss (3000 cycles): negligible. CA @ 10 mA/cm$^2$, 12 h: steady current density. |

CP denotes chronopotentiometry test; LSV denotes linear swept voltammetry; CA denotes chronoamperometry test.

## 6. Synthesis of Different MXene Morphologies

### 6.1. Multilayer and Few-Layer MXenes

The typical synthesis of MXenes always involves the etching of its MAX phase in the first step. HF-etched MXenes usually result in multi-layered, accordion-structured MXenes requiring an intercalation step with organic solvents such as DMSO and NMP, plus further exfoliation via sonication to obtain few-layer MXenes [194,195]. Alternatively, LiF + HCl-etched MXenes can contain a mixture of both multilayer and few layers. The few layers are usually found in the supernatant [22,196]. The etching process is often lengthy, requiring 12 h to several days to effectively etch the 'A' layers. HF also poses health and environmental risks [148,176]. Recently, Numan et al. [197] successfully fabricated delaminated Ti$_3$C$_2$T$_x$ via the microwave-assisted hydrothermal method with LiF + HCl as an etchant. Under those conditions, as illustrated in Figure 10, dissolution of LiF is facilitated owing to the vibration of water molecules and ions from direct heat transfer. Intercalation is sped-up due to Li$^+$ ion vibration. High-quality delaminated Ti$_3$C$_2$T$_x$ was obtained at a very short time of 2 h at 40 °C. Hence, this method offers a less hazardous pathway to preparing few-layer MXenes. Other steps are then taken to tailor the morphology of few-layer MXenes into porous, rolled, or crumpled structures.

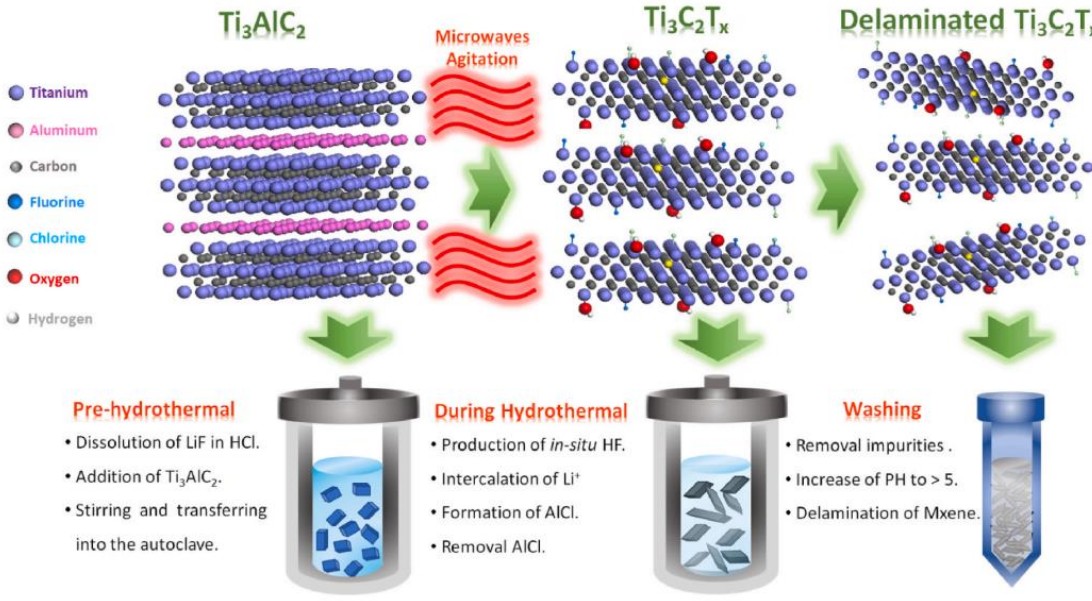

**Figure 10.** Schematic illustration of extraction of $Ti_3C_2T_x$ MXene from $Ti_3AlC_2$. Adapted with permission from [197], Copyright 2022, Elsevier.

*6.2. Preparation of Porous MXenes*

There are several ways to create porous MXenes, and these methods can produce a variety of MXenes with clearly defined porous architectures and porosity. Past reviews on porous MXene preparation have highlighted that coating on porous scaffolds, reassembly and crosslinking, template methods, in-plane pores fabrication and induced foaming are among the methods to obtain porous MXenes for applications such as in electromagnetic shielding, photocatalysis, sensors and energy storage [61,170]. Similar approaches were adopted in fabricating porous MXenes for HER applications.

6.2.1. Coating Porous Scaffold—Dip-Coating

Several scaffolds (or substrates) are dip-coated/deposited with 2D materials such as MXene to create electron bridging between conductive materials and HER electrocatalysts. Nickel foam (NF) is commonly selected as a porous scaffold to grow various electrocatalysts. NF is attractive as an HER substrate material given that it is corrosion-resistant, low weight, highly porous and possesses good electron conductivity. NF also participates in HER showing lower overpotentials in alkaline conditions than Cu foam or stainless steel mesh [198]. A multilayer or few-layer MXene colloidal solution of varying concentrations is prepared during coating. NF is then soaked into the MXene solution for a period of time before being dried. For instance, Li et al. [191] immersed the NF into a 4 mg/mL $Ti_3C_2T_x$ solution for 1 h to ensure the MXene was well-adsorbed onto NF. Drying was carried out under vacuum at mild temperature of 60 °C. It was then followed by a hydrothermal reaction to grow the electrocatalytically active material, in this case, FeCo-LDH, resulting in an interesting celosia-like morphology that participates in both HER and OER in alkaline conditions. Coating and direct growth on the porous scaffold would also create free-standing, binder-free electrodes that do not rely on binders. Binder-free electrodes are potentially more durable as they are able to overcome catalyst detachment issues due to unstable binders [199]. The concentration of MXene colloidal solution may vary from 0.1 to 6 mg/mL. Duration can change from 10 min to 2 h long [200,201]. Dipping frequency is also a factor, where it may form a layer-by-layer structure [202,203]. The thickness of the MXene layer on NF is controlled by varying these factors. Therefore, each factor must be carefully optimized for the desired HER properties in acid or alkaline conditions.

### 6.2.2. In-Plane Porous MXenes

The chemical etching method can introduce the pores/holes on the basal planes and edges of in-plane porous MXenes. Chemical etching is carried out in the presence of $H_2O_2$ and HCl over few-layer MXenes. Le et al. [182] used a chemical etching of $Ti_3C_2$ in a mixture of $HCl/H_2O_2$ with a volume ratio of 2/3 and a DI water fraction of 97 vol.%. Etching would be too rapid and non-homogenous if the DI water fraction were less than 97 vol.%. HCl facilitates pore formation. $H_2O_2$ oxidizes $Ti_3C_2$ to form $TiO_2$, while HCl dissolves the $TiO_2$ to create holes. Without HCl, the $TiO_2$ remains, and holes are absent. Figure 11 illustrates the steps of its synthesis. Hole size, BET surface area and pore volumes can be tuned by controlling etching time, where a longer time results in larger holes. However, the MXene sheets appear to be broken apart when the hole size is too large. IrCo supported on porous $Ti_3C_2$ chemically etched within 10 min showed the best OER activity. Besides HCl, HF can also be used, which may result in different properties of the holes [183]. Therefore, it may also be possible to carry out the etching of $TiO_2$ using other less hazardous acids. Other types of MXenes, such as $Mo_2C_3T_x$, may require different oxidating agents and acids for effective pore formation.

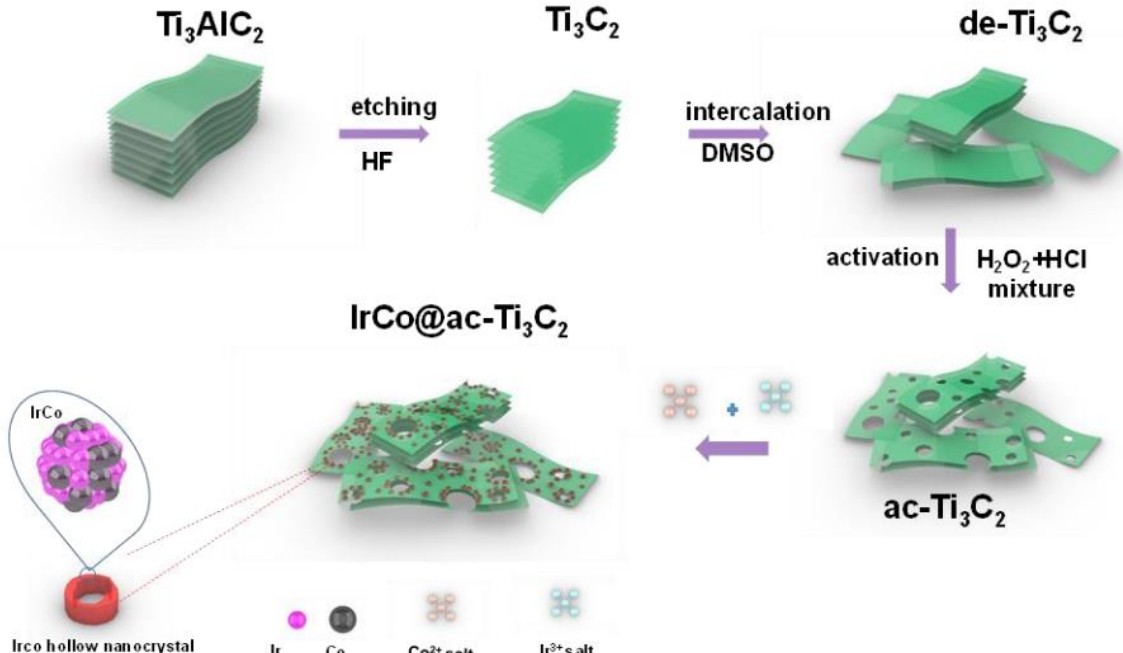

**Figure 11.** Scheme of the synthesis of IrCo supported on porous ac-$Ti_3C_2$. Adapted with permission from [182], Copyright 2020, John Wiley and Sons.

Poring agents or templates are able to create holes via a similar mechanism where the $TiO_2$ is being etched by acid. Kong et al. [184] first prepared a stable suspension containing few-layer $Ti_3C_2T_x$, $Pt^{2+}$ ions and $Ni^{2+}$ ions in the presence of poly(vinylpyrrolidone) and benzoic acid in benzyl alcohol. The mixture was subjected to a hydrothermal reaction to obtain $Pt_3Ni$-$Ti_3C_2T_x$. The product was mixed at a 1:1 weight ratio with polystyrene spheres in DI water, followed by sonication. Some amount of the suspension was then dropped onto a glassy carbon electrode (GCE), where the coated GCE was then immersed in toluene to etch away the polystyrene spheres to obtain the porous $Pt_3Ni$-$Ti_3C_2T_x$. This method may not result in compositional losses on $Ti_3C_2T_x$, as no $TiO_2$ is formed. Other templates include MgO [193]. It might be likely to tune the hole sizes by using different diameters of templates. Yet, it should be noted that the templates are inert to HER and can interfere with electrocatalytic activity. Hence, they should be removed as much as possible during washing.

### 6.2.3. Self-Assembly of MXene Nanosheets

Gelation-pyrolysis is a method to induce the self-assembly of MXene nanosheets into a porous MXene framework. A crosslinking agent is involved in assisting the assembly. An example is positively charged melamine (H+ melamine). Lin et al. [185] synthesized $H^+$ melamine by the gradual addition of concentrated HCl into an ethanol-melamine mixture. $H^+$ melamine was obtained after repeated washing and drying. The H+ melamine was then mixed with delaminated $Ti_3C_2T_x$ in DI water. Another mixture of Ir precursor (source of Ir SACs) and thiourea was quickly added into the $H^+$ melamine-$Ti_3C_2T_x$ mixture, which was then freeze-dried for 72 h. The freeze-dried mixture then underwent pyrolysis to decompose the $H^+$ melamine. Positively charged H+ melamine reduces the electrostatic repulsion between negatively charged MXene nanosheets, assembling them into the 3D porous framework that possesses larger ECSA than pristine MXene. The properties of the 3D porous MXene framework can be adjusted by varying the amount of melamine. The framework is also able to support Pd, Pt, and Ru SACs [186].

In the case of combining MXene nanosheets with graphene and/or g-$C_3N_4$ into a 3D porous structure, the method involved is co-assembly. Hydrogels consisting of two or three components are first obtained via hydrothermal/solvothermal method. He et al. [158] first sonicated a mixture containing as-prepared few-layered $Ti_3C_2T_x$, GO, and g-$C_3N_4$, each at similar 2 mg/mL concentrations. The mixture was then subjected to hydrothermal conditions at 100 °C for 12 h. GO was reduced to RGO during the assembly process, and a 3D monolithic hydrogel was formed, which then underwent 5-day dialysis with pure water. The resulting porous MX/CN/RGO was freeze-dried to avoid agglomeration and preserve the porous structure. Rigid $Ti_3C_2T_x$ is found to lodge in the pores and spaces between g-$C_3N_4$ and RGO nanosheets, minimizing restacking. The amount of MXene should be appropriate, as too much MXene will loosen the connection between electrically conducting RGO, while too little MXene (or g-$C_3N_4$) limits the active sites [187].

### 6.3. Crumpled and Rolled MXenes

Reports are still limited in fabricating crumpled and rolled MXenes. For crumpled MXene, Wu et al. [171] spray-dried a colloidal solution containing a few-layer MXene at 200 °C with air as the carrier gas. Unlike freeze-dried MXene, which tends to agglomerate, the spray-dried MXene appeared to form a crumpled morphology due collapse of the 2D nanosheets when the solvent was evaporated during the spray-drying process. Prior to spray drying, the MXene solution and the HER-active phase, in this example Pt, are mixed together. On the other hand, rolled MXenes involve a rapid freeze-drying process [110,204]. Liu et al. [110] prepared $MoS_2$ supported on rolled MXenes by first mixing the precursor with a few-layer MXene solution. The mixture was rapidly freeze-dried in liquid nitrogen within a lyophilizer. Annealing was then carried out under $H_2$(20 vol.%)/Ar(80 vol.%). It is suggested that rapid freeze-drying in liquid nitrogen causes the MXene nanosheet to roll due to sudden changes in strain. $MoS_2$ (or other nanoparticles/nanosheets) may exert more localized strain on the MXene nanosheets, which may facilitate rolling. Nanoparticles/nanosheets on the crumpled or rolled MXenes, as well as appropriate annealing conditions in a reductive environment, do not change their structure. This indicates that the crumpled and rolled MXenes are stable to some extent.

Overall, the above methods are feasible in preparing MXenes with different morphologies that cater to HER application. A faster alternative to produce few-layer MXene is feasible with microwave assistance. Careful selection of poring agents and crosslinking agents, as well as optimizing the reaction conditions, enables the formation of porous structures with the desired BET surface area and sufficiently exposed active areas to enhance electrocatalytic HER activity. Spray drying and freeze drying under specific conditions for few-layer MXenes can produce crumpled or rolled MXene morphologies. Porous, crumpled and rolled MXenes are able to overcome the restacking issue of MXene sheets. Additionally, every MXene morphology is potential support for HER-active materials. Further optimization of fabrication conditions is essential to ensure the balance between durability and

HER activity of MXenes and MXene composites. The advantages and limitations of these synthesis methods are summarized in Table 4.

**Table 4.** Summary of the advantages and limitations of some known preparation methods of MXenes with different morphologies.

| Synthesis method | Morphology | Advantages | Limitations |
|---|---|---|---|
| HF-etching of MAX phase to MXene [194] | Multi-layered MXenes | • Effectively etches the 'A' site to form the MXene structure.<br>• Formation of defects on the Ti that can occupy a metal atom.<br>• Concentration of F-termination is adjustable by varying concentrations of HF. | • Use of hazardous acid.<br>• Time-consuming.<br>• Additional intercalation step is required to produce few-layer MXene. |
| LiF + HCl-etching of MAX phase to MXene [22,194] | Few-layered MXenes, multi-layered MXenes | • Li$^+$ ions act as intercalants to produce a few-layer MXene.<br>• Few layers can be further tailored to MXene with special morphologies. | • MXene colloidal and sediments contain a mixture of multilayer and few layers that need to be separated. |
| Microwave-assisted LiF + HCl-etching of MAX phase to MXene [197] | Few-layered MXenes | • Shorten the time to produce few-layer MXenes.<br>• Effective etching and intercalation with negligible traces of Al.<br>• Facilitate the dissolution of LiF in HCl. | • Not all MXene flakes are delaminated. |
| Coating MXene on porous scaffold (i.e.: NF) through dip-coating [181,201] | MXene on porous scaffold | • Well-distributed macropores of varying sizes.<br>• Good adhesion between MXene and scaffold facilitates electron conductivity. | • Appropriate coating/dipping conditions required to effectively adsorb MXene on scaffold. |
| Chemical etching of MXene in the presence of acid and H$_2$O$_2$ [170,182,183] | In-plane porous | • Well-distributed meso/micro/macropores of tailorable sizes.<br>• Layered/flake structure of the MXene can be maintained. | • Losses of some Ti from the formation of TiO. Some termination groups lost due to holes. |
| Poring agents/templating [170,184] | In-plane porous | • Well-distributed, tailorable pore shape and sizes based on the template properties. | • Losses of termination groups due to holes.<br>• Trace poring agents may affect the HER activity. Extensive purification may be needed. |
| Gelation-pyrolysis for self-assembled porous structure [170,205] | Self-assembled porous structure | • 3D porous structure of MXene with well-defined porosity.<br>• Tailorable pore size through changing the concentration of crosslinking agent. | • MXene sheets may restack during assembly. |
| Co-assembly for multicomponent porous structure [158,192,205] | Multicomponent self-assembled porous structure | • Enables construction of highly electron-conductive 3D networks.<br>• Co-assembly possible to be carried out in hydrothermal conditions at shortened period of 4–6 h. | • MXene sheets may restack, while graphene or g-C$_3$N$_4$ may aggregate during assembly. |
| Spray drying of colloidal few-layer MXene [171] | Crumpled MXene | • Forms porous, crumpled structure that does not change when HER-active material added.<br>• Air can be used as carrier gas during spray drying. | • May only be effective on specific thickness of MXene nanosheets. |
| Rapid freeze drying [110,204] | Rolled MXene | • Rolled structure does not change when HER-active material is introduced.<br>• Rolled structure is stable under reductive annealing conditions (T = 350 °C).<br>• Allow for vertical assembly of nanosheets (i.e., MoS$_2$). | • May only be effective on specific thickness of MXene nanosheets.<br>• Freeze-drying process to obtain rolled structure may take a week. |

## 7. Conclusions and Future Prospects

This review highlighted several transition-metal-based and different structured MXene-based HER electrocatalysts in electrocatalytic water splitting. Transition metal-based electrocatalysts such as Ni-Co-P and $CuCo_2S_4$ are potentially lower-cost HER electrocatalysts with excellent activity but still do not surpass those of PGM-based electrocatalysts. Utilizing 2D MXenes as support materials for these nanoparticles/nanosheet electrocatalysts or those in the form of SACs as composite/hybrid catalysts further improves HER in both acidic and basic conditions. MXenes of different morphologies, whether it is Ti-based or Mo-based, facilitate a more homogenous distribution of catalyst nanoparticles/nanosheets, accelerate electron conductivity and bring the $\Delta G_{Hads}$ to a near-optimal value to benefit HER. Well-synthesized and purified few-layer MXenes consisting of thin nanosheets enable more exposure of the termination groups. Porous MXenes with well-defined porosity ease the access of electrolyte materials to the HER-active phases. There is also a potential for the crumpled and rolled MXenes as supports. Crumpled MXenes exhibit a certain extent of porosity that enlarges surface area to accommodate HER-active phases and active site access, while MXene nanorolls may offer the vertical alignment of catalyst nanosheets that also facilitate active site access. Few active PGM-free MXene composites, such as the 3D porous structure of $Ti_3C_2T_x$-rGO-g-$C_3N_4$ displayed comparable HER performance to Pt SACs supported on a self-assembled MXene porous framework in 0.5 M $H_2SO_4$. Therefore, the careful selection, combination and optimization of active transition-metal-based HER electrocatalysts with an equally HER-active MXene of suitable morphology are one of the strategies to develop a high-performing non-PGM catalyst. However, challenges remain on the long-term durability of these catalysts under current densities greater than 10 mA/cm$^2$ and changing conditions. Improvement to the fabrication methods should also be considered in upcoming works.

HER electrocatalysts must be both active and durable for continuous $H_2$ generation. Operation of water electrolysers in extended duration to 10,000 h with high current densities and fluctuating sources requires highly stable catalysts that do not degrade under such operations. Nanoparticles tend to leach from the electrode under high or fluctuating currents. Future studies on durability require focusing on the long-term stability of various MXene structures and the MXene composites within these actual water electrolyser systems. The mechanical stability of MXenes can play a role in their long-term durability. It can be achieved through a combination of MXene with mechanically stable materials, such as graphene. Furthermore, there is also an opportunity to explore other MXenes besides those of $Ti_3C_2T_x$, such as $Mo_2CT_x$, as well as V, Sc, and Ge-based MXenes, and the possibilities of tuning their morphologies since they also showed promising HER activity in terms of their fabrication. It is recommended to investigate the utilization of less hazardous, lower-cost and environmentally friendly options. Each step needs careful optimization to control the morphology of MXenes as well as the resulting MXene composites. These are crucial in an effort to not only produce active and stable catalysts but also to bring down the already high cost of MXenes and for better upscale productions. The microwave-assisted approach offers a potentially faster, low-cost, and possibly easy to upscale option, whether in MXene synthesis or hybrid preparation. Another method, known as the surface acoustic waves approach, is an interesting method for rapid production of delaminated MXenes that requires further exploration. Etchants based on ionic liquids are one feasible option for an acid-free etchant. An alternative to ionic liquids, the deep eutectic solvent, can also be explored to utilize it as an etchant or as an intercalant in the MXene preparation.

**Author Contributions:** Conceptualization, W.Y.W. and M.K.; writing—original draft preparation, R.R.R.S. and A.H.; writing—review and editing, W.Y.W., V.C., R.W. and M.K.; visualization, R.R.R.S., A.H. and M.K.; supervision, W.Y.W., M.K., R.M.Y., K.S.L., R.W. and M.K.; project administration, W.YW. and M.K.; funding acquisition, W.Y.W. and M.K. All authors have read and agreed to the published version of the manuscript.

**Funding:** This research was funded by Universiti Kebangsaan Malaysia, grant number GUP-2022-080, and Sunway University, grant number STR-IRNGS-SET-GAMRG-01-2022.

**Data Availability Statement:** Not Available.

**Acknowledgments:** The authors thank Universiti Kebangsaan Malaysia and Sunway University for providing the necessary resources to support this project.

**Conflicts of Interest:** The authors declare no conflict of interest.

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
