# Peer review of "Structurally Modified MXenes-Based Catalysts for Application in Hydrogen Evolution Reaction: A Review"

_catalysts, doi:10.3390/catal12121576_

Round 1
Reviewer 1 Report
In this review, Khalid and co-authors provided a comprehensive overview of structurally modified MXene-based catalysts for electrocatalytic hydrogen evolution reaction (HER). Indeed, MXene is gaining increasing research attention as promising catalyst candidates toward the HER. The review is therefore a timely one. In addition, the review is well-organized, with a summary of HER fundamentals, state-of-the-art catalysts, and features of MXene materials and their recent developments and achievements in HER, with a special focus of different MXene morphologies. The published data were well summarized in tables which can offer interested readers a helpful reference and guidance. The manuscript was also well drafted in a concise manner. Overall, the review is a timely contribution and suits the readership of Catalysts. Based on the expertise of this reviewer, I would in general support publication. However, some technical issues need to be resolved before the possible acceptance of the manuscript. Please see below for more detail.
1. The keywords used in Web of Science to get the search results of Figure 2, and the access date of the web of science database, should be provided in the figure caption.
2. The review is about the application of MXene-based materials for HER electrocatalysis. However, the display items (figures in particular) did not present any HER data. Please try to include some.
3. The reference section can be improved. Very recent works on hydrogen energy, HER and transition metal oxide HER catalysts are suggested to be referenced (e.g., Materials Reports: Energy, 2022, DOI: 10.1016/j.matre.2022.100144; Energy Fuels 2021, 35, 13585; Energy Technology, 2022, 10, 2200573, DOI: 10.1002/ente.202200573; Adv Mater, 2016, 28, 6442).
4. The current manuscript focuses primarily on tuning morphology of MXene-based materials for improving HER catalysis while paid less attention to the electronic structure tuning. As also mentioned in the last section (“The synthesis should consider optimisation to obtain a proper electronic structure of MXenes”), it is suggested that the authors provide some discussion on tuning of electronic structure of MXenes in terms of increasing HER performance. In this case, the quality of the manuscript will be further improved.
5. Figure 3 and figure 4 (middle part figure), if the figures were not drafted by the authors, permission should be requested.
6. Please modify the subtitle of section “2.3. Catalyst activity for current-time curve (i-e., chronopotentiometry)”. Based on the content discussed in this section, the part in the bracket should actually be deleted. Please note the current-time curve should be referred to as chronoamperometry rather than chronopotentiometry.
Author Response
Thank you for your response and reviewers’ useful comments on our manuscript. We have modified the manuscript in response to the reviewer's comments. The detailed corrections and rebuttals in response to the comments are made point by point which is given below.
Response to Reviewer 1:
In this review, Khalid and co-authors provided a comprehensive overview of structurally modified MXene-based catalysts for electrocatalytic hydrogen evolution reaction (HER). Indeed, MXene is gaining increasing research attention as a promising catalyst candidate toward the HER. The review is therefore a timely one. In addition, the review is well-organized, with a summary of HER fundamentals, state-of-the-art catalysts, and features of MXene materials and their recent developments and achievements in HER, with a special focus of different MXene morphologies. The published data were well summarized in tables which can offer interested readers a helpful reference and guidance. The manuscript was also well drafted in a concise manner. Overall, the review is a timely contribution and suits the readership of Catalysts. Based on the expertise of this reviewer, I would in general support publication. However, some technical issues need to be resolved before the possible acceptance of the manuscript. Please see below for more detail.
- The keywords used in Web of Science to get the search results of Figure 2, and the access date of the web of science database, should be provided in the figure caption.
- The keywords used have been included in the figure caption (Page 4, lines 115-116)
- The review is about the application of MXene-based materials for HER electrocatalysis. However, the display items (figures in particular) did not present any HER data. Please try to include some.
- Additional figures have been added under Section 5 displaying the HER performance of some MXene-based catalysts. The following are Figure 6 (Section 5.1, Pg 21, Line 608) and Figure 7 (Section 5.2, Pg 23, Line 684). Figures have been renumbered.
- The reference section can be improved. Very recent works on hydrogen energy, HER and transition metal oxide HER catalysts are suggested to be referenced (e.g., Materials Reports: Energy, 2022, DOI: 10.1016/j.matre.2022.100144; Energy Fuels 2021, 35, 13585; Energy Technology, 2022, 10, 2200573, DOI: 10.1002/ente.202200573; Adv Mater, 2016, 28, 6442).
- More recent works have been cited in this manuscript and are highlighted in the Introduction section and References section. (Introduction, Page 2 Lines 48 and 69, Page 3 Lines 77 and 86).
- The current manuscript focuses primarily on tuning morphology of MXene-based materials for improving HER catalysis while paid less attention to the electronic structure tuning. As also mentioned in the last section (“The synthesis should consider optimisation to obtain a proper electronic structure of MXenes”), it is suggested that the authors provide some discussion on tuning of electronic structure of MXenes in terms of increasing HER performance. In this case, the quality of the manuscript will be further improved.
- The electronic structure of MXene with the discussion on tuning the electronic structure is provided under Section 4. The discussion considers electronic properties of non-modified MXene and the bonding formed between catalyst material with MXene affecting the electronic structure as well as HER properties are provided (Section 4, Pg 17 – 18, Line 488 – 525)
- Figure 3 and figure 4 (middle part figure), if the figures were not drafted by the authors, permission should be requested.
- We have requested permissions from publishers on all reused figures, to be used in this manuscript.
- Please modify the subtitle of section “2.3. Catalyst activity for current-time curve (i-e., chronopotentiometry)”. Based on the content discussed in this section, the part in the bracket should actually be deleted. Please note the current-time curve should be referred to as chronoamperometry rather than chronopotentiometry.
- We have amended section 2.3 title to “Catalyst activity for current-time curve” (removed the i.e. chronopotentiometry) to better present the content of this section.
Reviewer 2 Report
The paper adresses a review about Structurally Modified MXenes-based catalysts for Application in Hydrogen Evolution Reaction. However the research gap and need for this review is not clearly stated in the introduction.
Table 2 should be splitted in different tables for each subsection in state-of-art section 3. This way the performance could be compared within the same group of catalysts. The best alternatives could be later discussed.
Section 5 should include a table for the development of the properties according to the state-of-art.
Section 6: What are the advantages and drawbacks of the different synthesis methods?
Conclusions are too broad and they should provide more detailed recommendations for future research in this topic.
Author Response
Thank you for your feedback and useful comments on our manuscript. We have modified the manuscript in response to the comments. The detailed corrections and rebuttals in response to the comments are made point by point which are given below.
Response to Reviewer 2:
- The paper addresses a review about Structurally Modified MXenes-based catalysts for Application in Hydrogen Evolution Reaction. However, the research gap and need for this review is not clearly stated in the introduction.
- Research gap and need for the review were added under the Introduction which is on the need to identify the suitable structure and morphology of MXene for HER catalyst (Introduction, Pg 3, Line 98 – 104).
- Table 2 should be splitted in different tables for each subsection in state-of-art section 3. This way the performance could be compared within the same group of catalysts. The best alternatives could be later discussed.
- An additional Table 1 (pages 13 – 16) is added for the summary of PGM-based and transition metal-based catalysts properties for HER to be compared with the MXene-based catalysts. MXene-based catalysts are moved into a separate Table 3 (Section 5, Pg 28 – 32). We did not combine these two tables because the intention for Table 3 is to provide a comparison of the HER activity and durability based on the different morphologies of MXenes used as the HER active catalyst support. The suggestion of the MXene supported HER catalyst configuration are discussed in the summary under Section 5 (Page 26 – 27, Line 780 – 805) and conclusion and Future Perspective (Section 7, Page 40 – 41, Line 959 – 1005).
- Section 5 should include a table for the development of the properties according to the state-of-art.
- We thank the reviewer comment on this. However, we would like to apologize that we are unable to provide an additional table to this section as we believe that Table 3 which summarizes the HER properties of the HER active catalysts supported on various MXene types and morphologies is able to provide sufficient information to the readers to compare and select the suitable combination for further study. We hope that our intention is clarified to the reviewer.
- Section 6: What are the advantages and drawbacks of the different synthesis methods?
- A table of comparison between the advantages and drawbacks of the different synthesis methods is added to this section (Table 4, pages 38-39).
- Conclusions are too broad and they should provide more detailed recommendations for future research in this topic.
- The conclusion and future prospects have been rewritten with recommendations for future research (Section 7, Pg 39 – 41, Line 958 – 1005). We hope that it is now satisfactory.
Round 2
Reviewer 2 Report
The paper has significantly improved